# SGD Jittering: A Training Strategy for Robust and Accurate Model-Based Architectures

**Peimeng Guan** [1]  **Mark A. Davenport** [1]

## Abstract

Inverse problems aim to reconstruct unseen data from corrupted or perturbed measurements. While most work focuses on improving reconstruction quality, generalization accuracy and robustness are equally important, especially for safety-critical applications. Model-based architectures (MBAs), such as loop unrolling methods, are considered more interpretable and achieve better reconstructions. Empirical evidence suggests that MBAs are more robust to perturbations than black-box solvers, but the accuracy-robustness tradeoff in MBAs remains underexplored. In this work, we propose a simple yet effective training scheme for MBAs, called SGD jittering, which injects noise iteration-wise during reconstruction. We theoretically demonstrate that SGD jittering not only generalizes better than the standard mean squared error training but is also more robust to average-case attacks. We validate SGD jittering using denoising toy examples, seismic deconvolution, and single-coil MRI reconstruction. Both SGD jittering and its SPGD extension yield cleaner reconstructions for out-of-distribution data and demonstrates enhanced robustness against adversarial attacks.

## 1. Introduction

Inverse problems (IPs) aim to recover the underlying signal $x \in \mathbb{R}^n$ from corrupted observation $y \in \mathbb{R}^m$, where $m$ is often less than $n$, leading to non-unique solutions. IPs are typically modeled as:

$$y = Ax + z, \tag{1}$$

where $A \in \mathbb{R}^{m \times n}$ is a known forward model for linear IPs, and $z$ represents unknown noise in the observation.

Machine learning algorithms show significantly better reconstructions than classical optimization methods. Most of the work in the field aims to promote reconstruction quality using different deep-learning models, such as transformers (Fabian et al., 2022) and diffusion models (Whang et al., 2022). In particular, model-based architectures (MBAs) such as loop unrolling (LU) networks inspired by classical optimization methods, unroll the objective function into a sequence of trainable iterative steps with the forward model $A$ involved in every iteration. MBAs are considered to be more interpretable than neural networks that learn a direct inverse without using $A$ (Monga et al., 2021), and achieve state-of-the-art reconstructions in different fields, such as natural image restoration (Gilton et al., 2021) and medical image reconstruction (Bian, 2024).

While reconstruction quality is crucial, robustness and generalization accuracy are equally important for IPs. Robustness refers to the model's ability to handle unwanted noise, while generalization accuracy measures how well the model performs on unseen data. For example, in medical image reconstruction, a model must generalize well to detect subtle details like small bone fractures and be robust enough to eliminate artifacts caused by measurement noise. Despite their importance, robustness and generalization have not been thoroughly explored in the context of IPs, particularly in MBAs. While some studies provide empirical assessments of robustness (Gilton et al., 2021; Genzel et al., 2022), there is a noticeable lack of theoretical analysis (Gottschling et al., 2025).

In this work, we model MBA training as a bilevel optimization problem without making specific assumptions about neural network architectures. We theoretically analyze its generalization and robustness risks in solving denoising problems and demonstrate that standard mean squared error (MSE) training may perform poorly in both metrics.

Technologies like adversarial training (AT) (Fawzi et al., 2016; Gilmer et al., 2019; Kannan et al., 2018) and input noise injection (Fawzi et al., 2016; Gilmer et al., 2019; Kannan et al., 2018) are commonly used to enhance robustness, while approaches such as weight perturbation (Wu et al., 2020), stochastic training (Hardt et al., 2016) and sharpness-aware minimization (Foret et al., 2021) are de-

[1]Department of Electrical and Computer Engineering, Georgia Institute of Technology, Atlanta, USA. Correspondence to: M. D. <mdav@gatech.edu>.

*Proceedings of the 42nd International Conference on Machine Learning*, Vancouver, Canada. PMLR 267, 2025. Copyright 2025 by the author(s).

signed to improve generalization. However, there is often a tradeoff between robustness and accuracy (Tsipras et al., 2019; Zhang et al., 2023; Gottschling et al., 2025; Krainovic et al., 2023). Strategies like careful loss function design (Pang et al., 2022; Kannan et al., 2018; Zhang et al., 2019; Zheng et al., 2021), data augmentation (Zhang et al., 2018), and robust architectural choices such as dropout (Yang et al., 2020) and teacher-student models (Zhao et al., 2024) have been explored to mitigate this tradeoff in classification tasks, but the challenge of balancing accuracy and robustness in IP solvers still remains an open question (Gottschling et al., 2025). Recent work by Krainovic et al. (2023) introduces jittering in $y$ for black-box IP neural networks, achieving a comparable level of worst-case robustness to AT but with significant accuracy loss.

To bridge this gap, this work aims to improve robustness through a novel training scheme for MBAs, while maintaining generalization accuracy for IPs. Note that while specially designed architectures can also improve robustness or generalization, they are not discussed here as this work focuses on investigating different training schemes.

**Our Contributions**  In this work, we propose a stochastic gradient descent (SGD) jittering scheme in training MBAs. This method unrolls the objective function for solving IPs using a gradient descent algorithm, with small, random noises added to the gradient updates at each iteration to improve the training process. During inference, these jittering noises are removed, allowing the model to perform as it would in standard evaluation settings. This technique has demonstrated greater robustness and better generalization than traditional MSE training, while also being more time-efficient than AT. Our contribution can be summarized as follows,

- *Network Assumptions*: We investigate the performance of MBAs with general non-convex neural networks, where training often results in non-unique parameters. This analysis is crucial for demonstrating improvements across various metrics, particularly in robustness and generalization accuracy.

- *Robustness*: We theoretically prove that SGD jittering implicitly promotes average-case robustness than regular MSE training. In the experiments, we further demonstrate the effectiveness of SGD jittering under worst-case attacks.

- *Generalization*: To the best of our knowledge, this is the first work to formally analyze generalization accuracy in inverse problems under small perturbations in the test data, by reformulating the generalization risk specific to inverse problems. This goes beyond the commonly observed heuristic that a flatter loss landscape implies better generalization.

## 2. Related Work

**Robustness and Generalization for IPs**  Several strategies have been proposed to improve robustness in solving inverse problems, including training-time exposure to diverse perturbations such as noise injection and data augmentation (Krainovic et al., 2023; Zhou et al., 2019), and adversarial training (Calivá et al., 2021), enabling models to better handle noise during inference. Other approaches focus on architectural innovations—for example, diffusion models have shown inherent robustness to noise in MRI reconstruction (Güngör et al., 2023), while PINN (Peng et al., 2022) embed domain knowledge to enhance stability without compromising interpretability. To improve generalization in IPs, methods like data augmentation with synthetic perturbations (Guan et al., 2024) and domain adaptation via techniques such as CycleGAN (Zhu et al., 2017) have been used to expand the training distribution and improve adaptability to unseen data. Additionally, incorporating geometric constraints, as in (Jiang et al., 2022) for electrocardiographic image reconstruction, has shown improved generalization by embedding prior knowledge into the learning process. While these approaches typically target either robustness or generalization, achieving both simultaneously remains an open and challenging problem.

**Analysis of Robustness**  Smoothness is closely related to the robustness of neural networks. Works like Salman et al. (2019); Cohen et al. (2019); Lecuyer et al. (2019); Li et al. (2018) show that regularization in smoothness promotes the robustness of classifiers. On the other hand, the analysis of IP is very different from classifications. Krainovic et al. (2023) analyzes the worst-case robustness for IPs using a simple linear reconstruction model and directly solves for robustness risk. However, optimizing a convex model results in a solution that excels in only one metric (accuracy or robustness). In this work, we explore nonlinear MBAs in their general forms, demonstrating improvements across both metrics.

**Analysis of Generalization**  Demonstrating better generalization of classifiers often involves showing larger classification margins (Cao et al., 2019; Kawaguchi et al., 2022; Lim et al., 2021), but the proof for IPs is very different. A common but indirect approach shows an algorithm can regularize the Hessian of loss landscape with respect to the network parameters. Models that converge to flat minima in the loss landscape tend to generalize better in practice (Jiang* et al., 2020; Keskar et al., 2017; Neyshabur et al., 2017; Foret et al., 2021), although the precise theoretical mechanism underlying this phenomenon remains unclear. Many studies have shown that regularizing the Hessian of the loss landscape (Xie et al., 2021; Keskar et al., 2017; Foret et al., 2021) leads to flatter minima and have directly

linked this to reduced generalization error, but this connection still lacks comprehensive theoretical justification. In this work, rather than focusing on the connection to flat landscapes, we proposed a new generalization risk for IPs, and demonstrate that SGD jittering results in a smaller generalization risk than regular MSE training.

**Noise Injections**  Noise injection serves as an implicit regularization technique during training to enhance model robustness and generalization. When applied to input data, it can be viewed as a data augmentation strategy. This approach has been extensively studied in classification tasks (Fawzi et al., 2016; Gilmer et al., 2019; Rusak et al., 2020) and has recently demonstrated effectiveness in improving robustness for IPs (Krainovic et al., 2023). Layer-wise noise injection (Lim et al., 2021) in a recurrent network has been shown to generalize better for classifiers. Weight injection perturbs network parameters during training, which is shown to help escape from local minima (Zhu et al., 2019; Nguyen et al., 2019) and promote smooth solutions (Orvieto et al., 2023; Liu et al., 2021; Orvieto et al., 2022). Weights can also be perturbed before parameter updates to enhance generalization, as in stochastic gradient descent (SGD). We will explore SGD in detail in the next subsection. In a similar vein, Renaud et al. (2024) adds noise to plug-and-play algorithms during evaluation to enhance in-distribution performance, though it does not address its impact on robustness and generalization accuracy.

**Connection to SGD Training**  SGD training is effective in promoting generalization through implicit regularization (Smith et al., 2021). In large overparameterized models with non-unique global solutions, SGD favors networks with flat minima (Hochreiter & Schmidhuber, 1997). The implicit regularization properties of SGD have been explored in contexts such as linear regression (Zou et al., 2021), logistic regression (Ji & Telgarsky, 2019), and convolutional networks (Gunasekar et al., 2018). Unlike SGD training, which adds noise to network parameters, the proposed SGD jittering method introduces noise at each iterative update of the reconstructions in the lower-level optimization. Despite its name, SGD jittering is structurally similar to layer-wise noise injection within MBAs.

## 3. Definition of Robustness and Generalization

We formally define average-case robustness and generalization risks for a learned inverse mapping $H_\theta : \boldsymbol{y} \mapsto \boldsymbol{x}$ in IPs. Average-case robustness is also known as the "natural" robustness for average performance (Rice et al., 2021; Hendrycks & Dietterich, 2019), which is equally important as in worst-case. During evaluation, a small vector $\boldsymbol{g}$ sampled from the distribution $\mathcal{P}_g$ is added to a data point $\boldsymbol{x} \in \mathcal{D}$. The new ground-truth becomes $\boldsymbol{x}_g = \boldsymbol{x} + \boldsymbol{g}$, with

the corresponding observation $\boldsymbol{y}_g = \boldsymbol{A}\boldsymbol{x}_g + \boldsymbol{z} = \boldsymbol{y} + \boldsymbol{A}\boldsymbol{g}$.

**Definition 3.1.** The **average-case robustness risk** of $H_\theta$ measures the distance between the unperturbed $\boldsymbol{x}$ to the recovered signal $\hat{\boldsymbol{x}}_g = H_\theta(\boldsymbol{y}_g)$ with the presence of $\boldsymbol{g}$,

$$R_e(\theta) = \mathbb{E}_{\boldsymbol{x}, \boldsymbol{y} \sim \mathcal{D}, \boldsymbol{g} \sim \mathcal{P}_g} ||\boldsymbol{x} - H_\theta(\boldsymbol{y}_g)||_2^2. \quad (2)$$

**Definition 3.2.** The **generalization risk** of $H_\theta$ measures the distance between the actual ground-truth $\boldsymbol{x}_g$ to the recovered signal $\hat{\boldsymbol{x}}_g = H_\theta(\boldsymbol{y}_g)$ with the presence of $\boldsymbol{g}$,

$$\mathcal{G}(\theta) = \mathbb{E}_{\boldsymbol{x}, \boldsymbol{y} \sim \mathcal{D}, \boldsymbol{g} \sim \mathcal{P}_g} ||\boldsymbol{x}_g - H_\theta(\boldsymbol{y}_g)||_2^2. \quad (3)$$

Robustness measures how close the reconstruction is compared to the data within the dataset, and the generalization accuracy measures how well the reconstruction obeys the physics of the IP even for out-of-distribution (OOD) data. Defining generalization in regression is not straightforward, but for IPs, we can achieve this using the underlying forward models. Typically, a tradeoff between robustness and accuracy exists for classical feedforward neural networks (Zhang et al., 2019; Yang et al., 2020; Stutz et al., 2019). We aim to illustrate that the SGD jittering technique enhances robustness and accuracy for MBAs without incorporating explicit regularization.

## 4. Model-based Architecture Overview

MBAs draw inspiration from classical optimization techniques, where the underlying signal $\boldsymbol{x}$ is estimated via,

$$\hat{\boldsymbol{x}} = \arg\min_{\boldsymbol{x}} \frac{1}{2} ||\boldsymbol{y} - \boldsymbol{A}\boldsymbol{x}||_2^2 + r(\boldsymbol{x}),$$

where $r$ is an arbitrary regularization function to ensure the uniqueness of the reconstruction. When $r$ is differentiable, the optimal solution can be solved via gradient descent (GD) algorithm, iterating through steps $k = 1, 2, 3...$ with a constant step-size $\eta$,

$$\boldsymbol{x}_{k+1} = \boldsymbol{x}_k - \eta \boldsymbol{A}^\top (\boldsymbol{A}\boldsymbol{x}_k - \boldsymbol{y}) - \eta \nabla r(\boldsymbol{x}_k). \quad (4)$$

In a *loop unrolling* (LU) architecture using GD, the gradient $\nabla r$ is replaced by a neural network, denoted by $f_\theta$. The process begins with an initial estimate $\boldsymbol{x}_0$, which is iteratively updated using (4) with a fixed number of iterations, denoted by $K$. The final output $\boldsymbol{x}_K$ represents the predicted reconstruction and is compared to the desired ground-truth. The weights $\theta$ are then updated through end-to-end back-propagation. Different variants of LU are summarized in Monga et al. (2021).

## 5. MBA Training Scheme Overview

In this section, we summarize some common training schemes for general regression problems, adapted here for MBA training. Following this, we introduce the proposed SGD jittering method.

**Regular MSE Training** Mean-squared error (MSE) is the most commonly used training loss for regression and IPs. In particular, the goal of training a MBA is to minimize the following risk w.r.t. $\theta$,

$$R(\theta) = \mathbb{E}_{\boldsymbol{x},\boldsymbol{y}\sim\mathcal{D}}||\boldsymbol{x} - \hat{\boldsymbol{x}}||_2^2 \quad \text{where,}$$
$$\hat{\boldsymbol{x}} = \arg\min_x \frac{1}{2}||\boldsymbol{y} - \boldsymbol{A}\boldsymbol{x}||_2^2 + r_\theta(\boldsymbol{x}) \tag{5}$$

Notice that MBAs do not directly learn $r$, but instead learn its gradient $\nabla r$ using a neural network $f_\theta$. We denote $r_\theta$ to highlight its dependency of $\theta$. The lower-level optimization is implemented using iterative steps, where each iteration corresponding to a single module in a MBA. For LU architecture, with a finite and fixed number of iterations $K$, MSE training minimizes the squared distance between the output $\boldsymbol{x}_K$ and the ground-truth $\boldsymbol{x}$. However, this training scheme can overfit to patterns in training data, potentially making the model more vulnerable to variations in $\boldsymbol{y}$.

**Adversarial Training** AT is widely recognized as one of the most effective methods for training robust neural networks (Wang et al., 2019). AT finds the worst-case attack for each training instance and adds it to the input data designed to deceive the model. By learning from these challenging examples, AT improves the model's robustness. For MBA training, AT aims to minimize the worst-case robustness risk as follows,

$$R_\epsilon(\theta) = \mathbb{E}_{\boldsymbol{x},\boldsymbol{y}\sim\mathcal{D}} \begin{bmatrix} \max_{\boldsymbol{e}:||\boldsymbol{e}||_2\leq\epsilon} ||\boldsymbol{x} - \hat{\boldsymbol{x}}||_2^2 \\ \text{s.t. } \hat{\boldsymbol{x}} = \arg\min_x \frac{1}{2}||\boldsymbol{y} + \boldsymbol{e} - \boldsymbol{A}\boldsymbol{x}||_2^2 + r_\theta(\boldsymbol{x}) \end{bmatrix} \tag{6}$$

Fast gradient sign method and projected gradient descent (ProjGD) are common methods to find the attack vector $\boldsymbol{e}$, where Athalye et al. (2018) demonstrates that ProjGD adversarial attacks are more robust. However, AT is slow as it requires iterative solvers for the attack vector (Shafahi et al., 2019).

AT might suffer from poor generalizations. Imagine in noiseless IPs where $\boldsymbol{y} = \boldsymbol{A}\boldsymbol{x}$, and assume the inverse $\boldsymbol{A}^{-1}$ exists. The goal of AT is to learn an inverse mapping $H_\theta$ such that $H_\theta(\boldsymbol{y} + \boldsymbol{e})$ is close to $\boldsymbol{x}$, for non-zero vector $\boldsymbol{e}$. However, the true estimated is $\boldsymbol{A}^{-1}(\boldsymbol{y} + \boldsymbol{e}) = \boldsymbol{x} + \boldsymbol{A}^{-1}\boldsymbol{e}$. AT learns to ignore the latter term, thus introducing errors that can degrade generalization accuracy.

**Input Jittering** Noise injection to the input is widely used in general ML tasks to promote robustness (Salman et al., 2019; Cohen et al., 2019; Lecuyer et al., 2019). In IPs that aim to map from $\boldsymbol{y}$ to $\boldsymbol{x}$, Krainovic et al. (2023) proposes adding a zero-mean random Gaussian vector $\boldsymbol{w}$ to $\boldsymbol{y}$ as input to a neural network that learns a direct reconstruction. They demonstrate that, with careful selection of the jittering variance, this training scheme can attain a comparable level of worst-case robustness to that achieved by AT. In MBAs,

the input jittering approach minimizes,

$$J_{\sigma_w}^I(\theta) = \mathbb{E}_{\boldsymbol{w},(\boldsymbol{x},\boldsymbol{y})\sim\mathcal{D}}||\boldsymbol{x} - \hat{\boldsymbol{x}}||_2^2 \quad \text{where,}$$
$$\hat{\boldsymbol{x}} = \arg\min_x \frac{1}{2}||\boldsymbol{y} + \boldsymbol{w} - \boldsymbol{A}\boldsymbol{x}||_2^2 + r_\theta(\boldsymbol{x}). \tag{7}$$

This method learns an inverse mapping from $\boldsymbol{y} + \boldsymbol{w}$ to $\boldsymbol{x}$, which might suffer from generalization issues in the same way as AT. In fact, Krainovic et al. (2023) demonstrates a significant accuracy loss using input jittering for black-box neural networks.

# 6. Proposed Training Scheme

Another line of work injects noises layer-wise into the neural network, where Hodgkinson et al. (2021); Jim et al. (1996); Liu et al. (2020) model this layer-wise injection as a stochastic differential equation, demonstrating its role in implicit regularization for robustness. Additionally, Lim et al. (2021) introduces noise into recurrent neural networks and shows improved generalization for classification tasks. In this work, we propose adding random zero-mean Gaussian noises, $\boldsymbol{w}_k$, to the gradient updates $f_\theta(\boldsymbol{x}_k)$, with the noise re-sampled independently for each iteration $k = 1, 2, 3, ...$. The noisy gradients are unbiased, i.e., $\mathbb{E}[f_\theta(\boldsymbol{x}) + \boldsymbol{w}_k] = \mathbb{E}[f_\theta(\boldsymbol{x})]$ for all $\boldsymbol{x}$. Let $\bar{w} = \{\boldsymbol{w}_1, ..., \boldsymbol{w}_K\}$. The estimated reconstruction is solved iteratively using this noisy gradient descent, which is essentially stochastic gradient descent (SGD). Thus, with step size $\eta$, the learning objective to minimize the following risk,

$$J_{\sigma_{w_k}}^{SGD}(\theta) = \mathbb{E}_{\bar{w},(\boldsymbol{x},\boldsymbol{y})\sim\mathcal{D}}||\boldsymbol{x} - \hat{\boldsymbol{x}}||_2^2 \quad \text{where,}$$
$$\hat{\boldsymbol{x}} = \lim_{k\to\infty} \boldsymbol{x}_k - \eta(\boldsymbol{A}^\top(\boldsymbol{A}\boldsymbol{x}_k - \boldsymbol{y}) + f_\theta(\boldsymbol{x}_k) + \boldsymbol{w}_k). \tag{8}$$

The jittering noise satisfy the following assumption.

**Assumption 6.1.** (SGD noises) Assume that at each gradient descent iteration $k = 1, 2, 3, ...$, noise $\boldsymbol{w}_k \sim \mathcal{N}(0, \sigma_{w_k}^2/n\mathbf{I})$ is independently sampled from an i.i.d. zero-mean Gaussian distribution, where $\mathbb{E}||\boldsymbol{w}_k||_2^2 = \sigma_{w_k}^2$.

In AT and input jittering training, the noise vectors are added to $\boldsymbol{y}$ throughout the entire inversion, introducing a bias to the reconstruction compared to the actual ground-truth. In contrast, the lower-level objective for SGD jittering is the same as in regular MSE training. Extensive research has explored the convergence properties of SGD in solving nonconvex optimization problems (Ghadimi & Lan, 2013; Li et al., 2024; Madden et al., 2024; Lei et al., 2019), which ensures a correct inverse mapping even with the presence of noise, thus preserves high accuracy in reconstruction.

We adapted the SGD convergence result in (Garrigos & Gower, 2023) for SGD jittering as follows, where the proof of Corollary 6.2 can be found in the Appendix.

**Corollary 6.2.** (*Convergence of MBAs-SGD for IPs, modified from Theorem 5.12 in (Garrigos & Gower, 2023)) Let*

$F(\boldsymbol{x}) = \frac{1}{2}||\boldsymbol{y} - \boldsymbol{A}\boldsymbol{x}||_2^2 + r_\theta(\boldsymbol{x})$ *denote the lower-level objective function in (8). Assume* $f_\theta = \nabla r_\theta$ *is L-Lipschitz continuous, satisfying*

$$||f_\theta(\boldsymbol{x} + \Delta) - f_\theta(\boldsymbol{x})||_2 \leq L||\Delta||_2, \quad \forall \boldsymbol{x}, \Delta,$$

*Consider a sequence* $\{\boldsymbol{x}_k^{sgd}\}_{k=0}^K$ *generated by SGD in (8) with a constant step-size* $\eta = \sqrt{\frac{2}{LL_{\max}K}}$, *for* $K \geq 1$, *the following holds:*

$$\min_{0 \leq k < K} \mathbb{E}||\nabla F(\boldsymbol{x}_k^{sgd})||_2^2 \leq \sqrt{\frac{2LL_{\max}}{K}}\big(2(F(\boldsymbol{x}_0^{sgd}) - \inf F) + \Delta_F^*\big),$$

*where,* $\boldsymbol{x}^* = \arg\min_x F(\boldsymbol{x})$, *and*

$$\Delta_F^* = F(\boldsymbol{x}^*) - \inf\big(\frac{1}{2}||\boldsymbol{y} - \boldsymbol{A}\boldsymbol{x}||_2^2\big) - \inf r_\theta(\boldsymbol{x}),$$

*denote* $\lambda_i$ *as the eigenvalues of* $\boldsymbol{A}^\top \boldsymbol{A}$ *and* $L_{\max} = \max\{L, \max_{i=1,..,n} \lambda_i\}$.

On the other hand, SGD jittering noise also implicitly enhances robustness. While we will analyze this relationship theoretically in the following section, intuitively, exposing a neural network to noisier inputs during training encourages it to converge to smoother regions of the loss landscape, which are less sensitive to input variations. This results in more stable outputs that are less affected by perturbations during evaluation.

# 7. Analysis for Denoising Problem

In this section, we analyze the generalization and robustness of SGD jittering in addressing the denoising problem. We begin by outlining the relevant assumptions.

**Assumption 7.1.** (Denoising) The forward model in denoising is identity mapping, so that $\boldsymbol{y} = \boldsymbol{x} + \boldsymbol{z}$. Signals $\boldsymbol{x} \in \mathbb{R}^n$ are sampled from distribution $\mathcal{X}$, and noises are sampled i.i.d. from $\boldsymbol{z} \sim \mathcal{N}(0, \sigma_z^2/n\boldsymbol{I})$.

**Assumption 7.2.** (Neural Network) Assume the neural network architecture is twice differentiable with respect to the input, so the Hessian exists.

**Definition 7.3.** Let $\theta^{mse}$, $\theta^{at}$ and $\theta^{sgd}$ denote the optimal neural network parameters of $f$ obtained from regular MSE training in (5), AT in (6) and SGD jittering training in (8), respectively.

## 7.1. Main Theoretical Results

Given the assumptions, we state the first theorem regarding generalization accuracy. Let $H_\theta$ denote the inverse process learned using MBAs for the rest of the analysis.

**Theorem 7.4.** *Assuming a small zero-mean Gaussian random vector* $\boldsymbol{g}$ *with* $\mathbb{E}||\boldsymbol{g}||_2^2 = \sigma_g^2$ *is added to data* $\boldsymbol{x} \in \mathcal{D}$ *during evaluation. The generalization risk for denoising problem is defined as,*

$$\mathcal{G}(\theta) = \mathbb{E}_{\boldsymbol{x}, \boldsymbol{y} \sim \mathcal{D}, \boldsymbol{g}}||\boldsymbol{x} + \boldsymbol{g} - H_\theta(\boldsymbol{y} + \boldsymbol{g})||_2^2. \quad (9)$$

*Under assumptions 6.1, 7.1 and 7.2, SGD jittering generalizes better than regular MSE training,*

$$\mathcal{G}(\theta^{sgd}) \leq \mathcal{G}(\theta^{mse}).$$

The detailed proof is provided in Appendix C. SGD jittering can be interpreted as a noisy version of the regular MSE training. The idea is to decompose the generalization accuracy in the presence of $\boldsymbol{g}$ into two terms: the MSE term corresponding to the reconstruction from the clean trajectory $H_\theta(\boldsymbol{y})$, and a penalty term depends on the noisy trajectory $H_\theta(\boldsymbol{y} + \boldsymbol{g})$. Then we show that the training loss of SGD jittering includes an implicit regularization term that penalizes deviations in the intermediate reconstructions caused by the perturbation. Minimizing the SGD jittering training loss also minimizes the penalty term in $\mathcal{G}(\theta)$, leading to better generalization accuracy compared to regular MSE training.

Furthermore, we rewrite the regularization term derived in the proof as follows in SGD training via Taylor expansion.

$$\text{regularization} = \mathbb{E}\,||\sum_{i=0}^{K-1} \eta(1-\eta)^{K-1-i}(f_\theta(\boldsymbol{x}_i') - f_\theta(\boldsymbol{x}_i^{sgd}))||_2^2. \quad (10)$$

Let $\{\boldsymbol{x}_k^{sgd}\}_{k=0}^K$ and $\{\boldsymbol{x}_k'\}_{k=0}^K$ denote the reconstruction trajectory from SGD and GD, respectively. As derived through iterative expansions in the proof, at iteration $k$, we have $\boldsymbol{x}_k^{sgd} = \boldsymbol{x}_k' + \boldsymbol{\delta}_k$, where $\boldsymbol{\delta}_k = \sum_{i=0}^{k-1} \eta(1-\eta)^{k-1-i}(f_\theta(\boldsymbol{x}_i') - f_\theta(\boldsymbol{x}_i^{sgd})) + \sum_{i=0}^{k} \eta(1-\eta)^{k-i}\boldsymbol{w}_i$. Assumption 7.2 ensures the existence of the first and second derivatives of $f_\theta$. Let $f_\theta^i : \mathbb{R}^n \to \mathbb{R}$ be the function defined by $f_\theta^i(\boldsymbol{x}) = [f_\theta(\boldsymbol{x})]_i$, representing the $i^{th}$ component of $f_\theta(\boldsymbol{x})$, for $i \in \{1, ..., n\}$. Then, we have,

$$f_\theta^i(\boldsymbol{x}_k^{sgd}) \approx f_\theta^i(\boldsymbol{x}_k') + \boldsymbol{\delta}_k^\top \nabla f_\theta^i(\boldsymbol{x}_k') + \frac{1}{2}\boldsymbol{\delta}_k^\top \boldsymbol{H}_f^i(\boldsymbol{x}_k')\boldsymbol{\delta}_k,$$

where, $\nabla f_\theta^i$ and $\boldsymbol{H}_f^i$ denote the gradient and the Hessian of $f_\theta^i$. As $k$ increases, the regularization weight $\eta(1-\eta)^{K-1-i}$ in (10) becomes more significant. Thus minimizing the SGD jittering risk implicitly promotes smoother and wider landscapes of the lower-level objective $F$ with respect to the iterative inputs $\boldsymbol{x}_k'$ for larger values of $K$. It is important to note that this differs from the concept of flat minima in a general feedforward neural network, which flatness refers to regions of low training loss with respect to the network parameters, often associated with improved generalization. Our findings align with the observations in (Lim et al., 2021), which demonstrate that layer-wise noise injections to recurrent neural networks promotes a smaller Hessian with respect to the hidden inputs to the final layer. While it is widely observed that flat minima in the loss landscape (with respect to network parameters) imply better

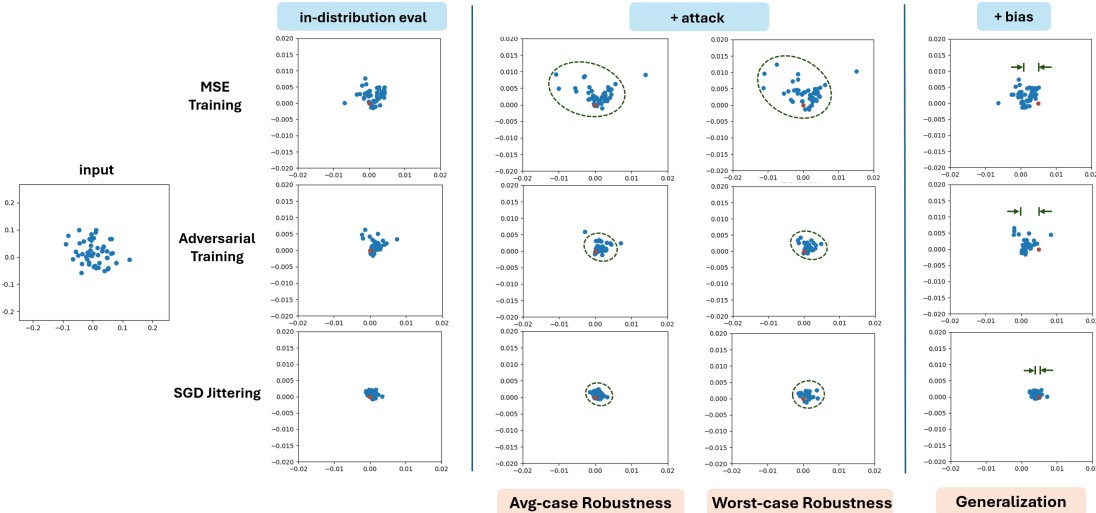

*Figure 1.* 2D point denoising example using MSE training, AT, and SGD jittering. Each method is evaluated using regular in-distribution data, under average- and worst-case attacks to $\boldsymbol{y}$ and with additional bias (from left to right columns respectively). Blue dots represents each test data instance, and the red dots indicate the ground-truth. The dashed circle highlights the cluster under attack, and the arrows represent the distance from the center of the data cluster to the biased ground truth.

generalization as summarized in Section 2, Lim et al. (2021) does not clarify why flat minima with respect to hidden-layer inputs improve generalization in regression settings. Our work fills this gap in the context of IPs.

Furthermore, we present the robustness analysis results.

**Theorem 7.5.** *Let $\boldsymbol{e}$ be the perturbation vector sampled i.i.d. from the zero-mean Gaussian distribution, $\mathcal{P}_e$, such that $\mathbb{E}\|\boldsymbol{e}\|_2^2 = \sigma_{\boldsymbol{e}}^2$. For an inverse mapping $H_\theta$, the average-case robustness risk for denoising is as follows,*

$$R_e(\theta) = \mathbb{E}_{(\boldsymbol{x},\boldsymbol{y})\sim\mathcal{D},\boldsymbol{e}\sim\mathcal{P}_e} \left[\|\boldsymbol{x} - H_\theta(\boldsymbol{y} + \boldsymbol{e})\|_2^2\right]. \quad (11)$$

*SGD jittering is more robust than MSE training against bounded-variance perturbations around the measurements,*

$$R_e(\theta^{sgd}) \leq R_e(\theta^{mse}).$$

The proof is similar to the one in proving generalization, we include the full proof in the Appendix D. The implicit regularization induced by SGD jittering help characterize both generalization and robustness of MBAs.

## 8. Experiments and Discussions

In our experiments, we use a toy denoising example to validate our theorems and demonstrate the proposed training scheme using seismic deconvolution and magnetic resonance imaging reconstruction. For each task, we train the same LU architecture using different training schemes. We then evaluate the models using in-distribution (ID) testing data, adversarial attacks with various strengths, and task-dependent OOD data. Detailed training procedures are provided in the Appendix.

**Toy Problem: 2-dimensional Point Denoising** We begin with a 2D point denoising problem to validate our theorems. The task involves recovering the point $\boldsymbol{x} = (0,0)^\top$ from noisy observations generated by adding zero-mean Gaussian noise with a variance of 0.01. We use 200 samples for training and 50 for evaluation. Figure 1 visualizes the denoising results under ID data, average-case attack, worst-case attack, and a biased ground-truth. ID data are sampled from the same distribution as the training data, centered at $(0,0)^\top$. The worst-case attack finds a vector within a 0.01 $\ell_2$-distance around $\boldsymbol{y}$ that maximizes the squared error, while the average-case attack uniformly samples attack vectors within the same ball and adds to $\boldsymbol{y}$. Finally, we evaluate generalization accuracy by reconstructing the slightly biased ground truth $(0.005, 0)^\top$ from its corresponding measurements. The results show that both SGD jittering and AT produce tighter clusters of reconstructed points compared to MSE training under adversarial attacks. Moreover, when reconstructing a slightly biased ground truth, SGD jittering achieves the most accurate reconstruction, showcasing its better generalization capability to small variations in data. Notice that MSE training underperforms SGD jittering on ID data, likely because the noise injection in SGD jittering helps the model escape from local minima in training.

**Seismic Deconvolution** Seismic deconvolution is a crucial problem in geophysics. It aims to reverse the convolution effects in the received signal, which occurs when artificial source waves travel through the Earth's layers. The goal is to extract sparse reflectivity series from recorded seismic data. The training data is generated following the same procedure as in (Iqbal et al., 2019; Guan et al., 2024).

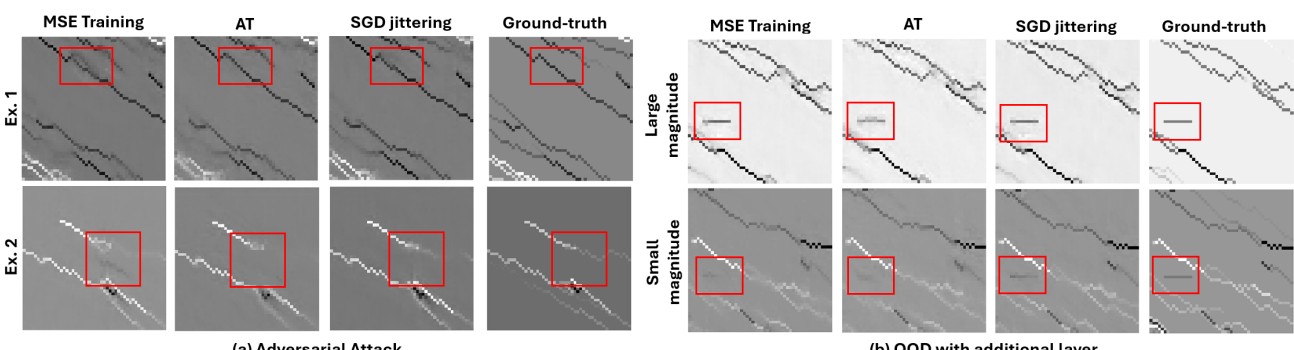

*Figure 2.* Visualization of seismic deconvolution using (stochastic) **gradient descent** variants within loop unrolling architectures. Figure (a) illustrates the reconstruction performance under adversarial attack with $\epsilon = 1$. Figure (b) evaluates the model's performance on OOD seismic data, where the ground-truth reflectivity includes an additional small horizontal layer (with magnitude of 0.3 and 0.1), highlighted in the red boxes.

| PSNR / SSIM | ID Test Data | Adv. Attack | OOD Data |
|---|---|---|---|
| MSE training | 34.64 / 0.921 | 28.85 / 0.829 | 34.57 / 0.918 |
| AT | 33.50 / 0.903 | **30.27** / **0.849** | 33.45 / 0.902 |
| Input Jittering | 32.92 / 0.882 | 30.10 / 0.832 | 32.91 / 0.882 |
| **SGD jittering** | **35.10** / **0.928** | 29.89 / 0.842 | **34.93** / **0.927** |

| PSNR / SSIM | fastMRI | Adv. Attack | Tumor Cell |
|---|---|---|---|
| MSE training | 28.21 / 0.603 | 25.68 / 0.382 | 29.92 / 0.779 |
| AT | 27.68 / 0.564 | **27.17** / 0.549 | 27.74 / 0.597 |
| Input Jittering | 28.18 / 0.595 | 25.05 / 0.420 | 29.97 / 0.740 |
| **SGD jittering** | **28.22** / **0.607** | 26.77 / **0.552** | **30.36** / **0.788** |

*Table 1.* **Seismic deconvolution** evaluation for in-distribution data (column 2), under adversarial attack with $\epsilon = 1$ (column 3), and for OOD data with additional horizontal layer (columns 4). The best and second-best performances are in bold and underlined respectively.

*Table 2.* **MRI evaluation** for in-distribution data (column 2), adversarial attack (column 3), and OOD data (column 4). The best and second best performances are in bold and underlined respectively.

To test the generalization accuracy, we generate OOD data by introducing an arbitrary horizontal layer with magnitudes of 0.1 to the ground truth, reflected accordingly in the measurements. The numerical results in Table 1 show that SGD jittering achieves the highest peak signal-to-noise ratio (PSNR) and structural similarity index (SSIM) on ID and OOD data, while maintaining competitive performance under adversarial attacks. In contrast, AT demonstrates the best robustness under adversarial attacks but suffers significant performance degradation for both ID and OOD data. Visual reconstructions in Figure 2 further highlight the differences between these methods. While MSE training often introduces artifacts, especially under adversarial attacks, AT fails to recover the small horizontal layer in OOD data with a magnitude of 0.1, likely perceiving it as noise. In comparison, SGD jittering accurately recovers the additional layer, demonstrating its ability to generalize to subtle features not present in the training data.

**Accelerated MRI Reconstruction** Accelerated MRI aims to recover human-interpreted body structures from partial k-space measurements. We train models with single-coil knee MRI from the fastMRI dataset (Knoll et al., 2020) with $4\times$ acceleration, or $1/4$ of the measurements in k-space is

used for reconstruction. To assess generalization, we use a different knee dataset from Bickle & Jin (2021), which includes giant tumor cells absent from the training data. The results in Table 2 show that, on ID data, MSE training and SGD jittering perform similarly, while AT underperforms due to its inherent tradeoff in resolution. On OOD tumor data, SGD jittering outperforms all other methods, demonstrating better generalization to unseen features. Visual comparisons in Figure 3 further support these findings. Reconstructions produced by AT are smoother but often miss fine details, consistent with prior observations in black-box IP solvers (Krainovic et al., 2023). In contrast, SGD jittering produces high-resolution reconstructions, preserving critical anatomical structures, particularly in OOD cases.

### 8.1. Robustness vs. Generalization

Figure 4 presents a comparison of various training schemes in terms of in-distribution accuracy, robustness and generalization measured by average test PSNR on seismic deconvolution (top row) and MRI reconstruction (bottom row) tasks. The left panels show that the proposed SGD jittering approach achieves better generalization to OOD data. Meanwhile, the right panels depict the trade-off between robustness and accuracy for baseline methods, where our

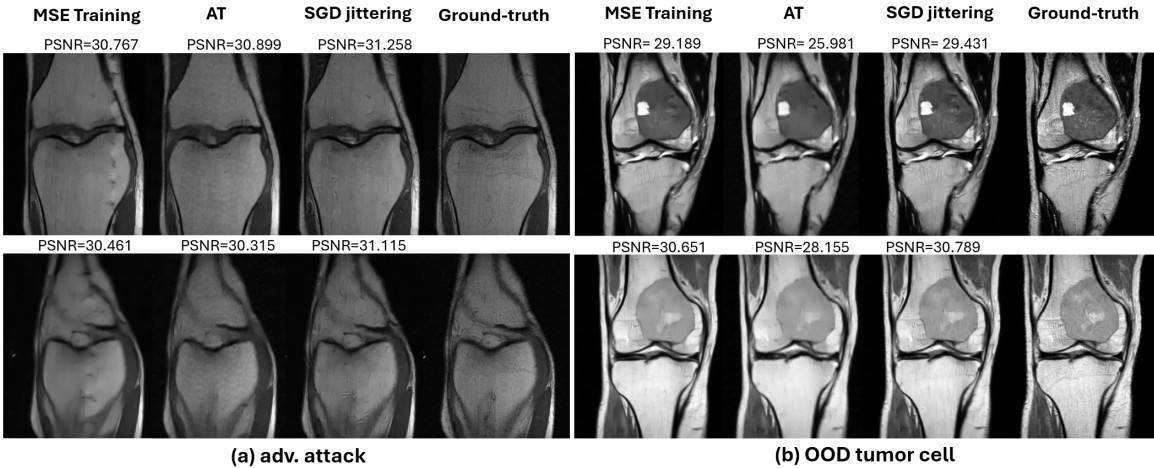

*Figure 3.* **Gradient descent** LU for MRI evaluation (a) under **adversarial attack** obtained from projected gradient descent with norm no more than $\epsilon = 1$. Each row represents an MRI reconstruction of a sample from fastMRI (Knoll et al., 2020). (b) MRI evaluation using **OOD** tumor knee data from (Bickle & Jin, 2021).

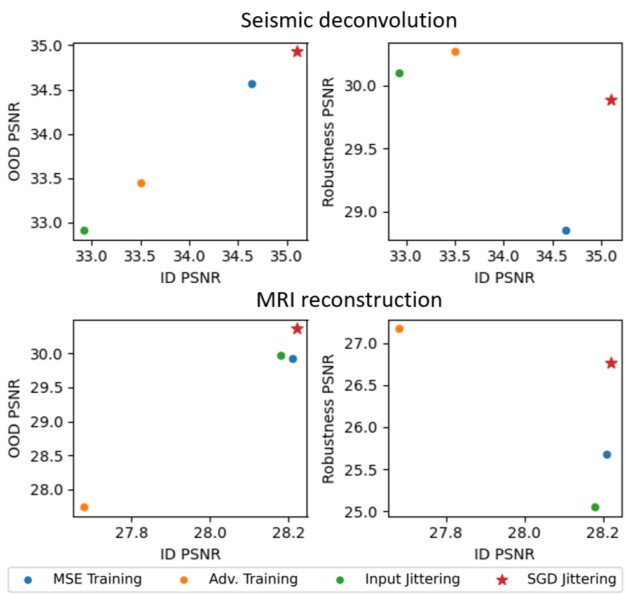

*Figure 4.* Comparing robustness and generalization of different training schemes for GD LU for seismic deconvolution problem (top row), and MRI reconstruction (bottom row).

approach provides a more effective balance between the two metrics.

### 8.2. How to Choose SGD Noise Level?

Figure 6 examines the impact of SGD jittering noise levels $\sigma_{wk}^2$ in the context of the denoising problem. The jittering noise level determines the robustness and in- and out-of-distribution accuracies. This parameter search allows the identification of an optimal $\sigma_{wk}^2$ that balances robustness against adversarial attacks while maintaining high accuracy.

### 8.3. Stochastic Proximal Gradient Descent Jittering

We further extend the proposed stochastic training idea to proximal gradient descent (PGD) variant of model-based architecture training, and denote the algorithm as SPGD jittering, where the learning objective is,

$$J_{\sigma_{w_k}}^{SPGD}(\theta) = \mathbb{E}_{\bar{\boldsymbol{w}},(\boldsymbol{x},\boldsymbol{y})\sim\mathcal{D}}||\boldsymbol{x} - \hat{\boldsymbol{x}}||_2^2 \quad \text{where,}$$
$$\hat{\boldsymbol{x}} = \lim_{k\to\infty} prox_\theta\big(\boldsymbol{x}_k - \eta\big(\boldsymbol{A}^\top(\boldsymbol{A}\boldsymbol{x}_k - \boldsymbol{y}) + \boldsymbol{w}_k\big)\big). \tag{12}$$

Here, $prox$ represents a proximal operator, which is replaced by a neural network parameterized by $\theta$, and $\boldsymbol{w}_k$ denotes the stochastic noises injected independently to the input of the proximal network at each iteration. Similar to SGD jittering, SPGD jittering ensures that the noise $\boldsymbol{w}_k$ is re-sampled at every iteration, maintaining unbiased gradient updates while promoting variations in training data.

Figure 5 compares the same *proximal gradient descent* framework with different training schemes for accelerated MRI reconstruction. MSE training denotes the classical proximal GD variant of LU trained with MSE loss, AT, and SPGD jittering for accelerated MRI reconstruction. MSE training demonstrates reasonable generalization to OOD tumor data but remains highly sensitive to adversarial attacks. In contrast, AT exhibits robustness against attacks but produces reconstructions with lower resolution, which can obscure fine details in critical medical imaging tasks. SPGD jittering effectively enhances both robustness and accuracy, achieving cleaner reconstructions under adversarial attacks and superior generalization to OOD tumor data.

Numerical evaluations on SPGD jittering for accelerated MRI and further experiments on seismic deconvolution are presented in Appendix E. We also compare the training speed of AT and the proposed stochastic noise injection

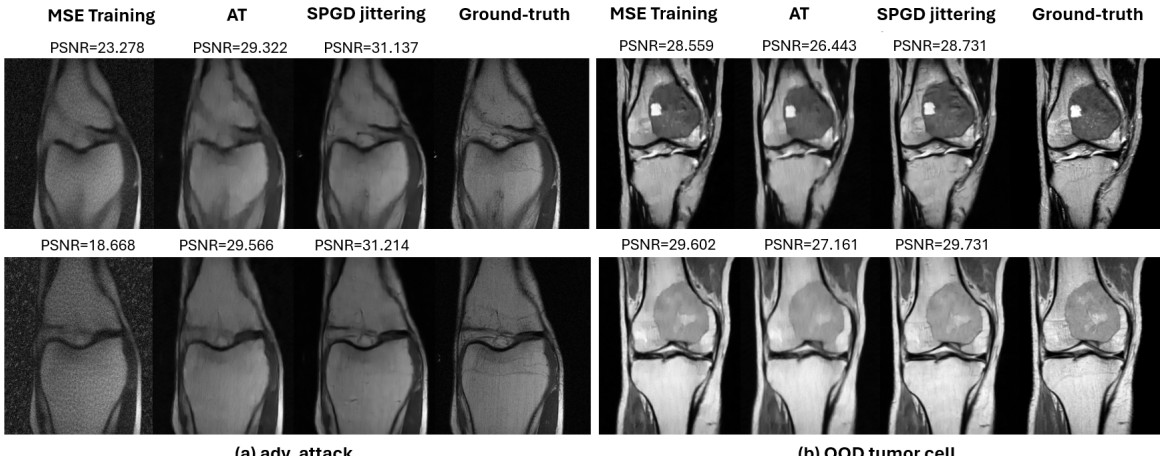

**(a) adv. attack**   **(b) OOD tumor cell**

*Figure 5.* **Proximal gradient descent** LU for MRI evaluation (a) under **adversarial attack** obtained from projected gradient descent with norm no more than $\epsilon = 1$. Each row represents an MRI reconstruction of a sample from fastMRI (Knoll et al., 2020). (b) MRI evaluation using **OOD** tumor knee data from (Bickle & Jin, 2021).

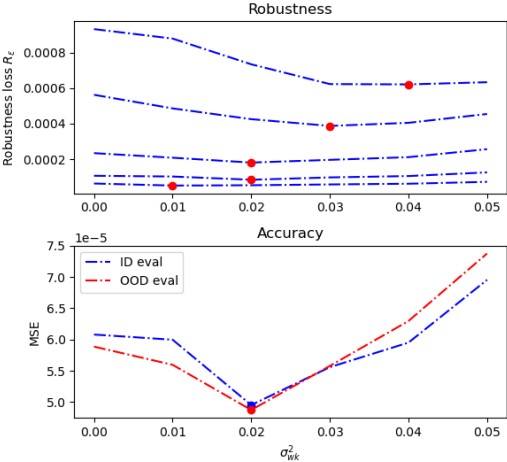

*Figure 6.* The effect of SGD jittering noise level $\sigma_{wk}^2$ on model performance for denoising problem. (Top): Worst-case robustness $R_\epsilon$ recorded for each noise level. Each line represents a different value of $\epsilon$, with the red dot indicating the $\sigma_{wk}^2$ that achieves the best robustness for each corresponding $\epsilon$. (Bottom): In-distribution (ID) and out-of-distribution (OOD) accuracies in MSE are recorded for each $\sigma_{wk}^2$, with best noise levels highlighted.

techniques in Appendix H, demonstrating that the proposed method is a time-efficient strategy to promote robustness.

## 9. Conclusion

Robustness and generalization accuracy are both crucial in solving IPs, yet current analyses often focus on simple black-box solvers that enhance robustness at the cost of accuracy. While empirical studies have examined the robustness of MBAs, there is a lack of theoretical analysis. In this work, we introduce a novel and easily implemented

SGD jittering training scheme to address the tradeoff between robustness and accuracy in solving IPs. Our method proves effective for high-quality model-based IP solvers. Through mathematical analysis, we demonstrate that this approach implicitly regularizes the gradient and Hessian of the neural network with respect to the input, resulting in improved generalization and enhanced robustness compared to traditional MSE training. Experimental results show that while MSE training is vulnerable to adversarial attacks and AT recovers smooth estimates, SGD and SPGD jittering consistently produces robust, high-quality outcomes.

## Software and Data

The code for the proposed SGD and SPGD jittering methods are provided here: https://github.com/InvProbs/SGD-jittering.

## Acknowledgements

This work was supported by the Center for Energy and Geo Processing (CeGP) at Georgia Institute of Technology.

## Impact Statement

This work highlights the critical need for both robustness and generalization accuracy in solving inverse problems. Model-based architectures are high-performance and data-efficient machine learning architecture for IPs. The proposed SGD/SPGD jittering strategy mitigates the robustness-accuracy tradeoff for this specific architecture, demonstrating both efficiency and effectiveness across diverse applications. However, for inverse problems with unknown forward models where model-based architectures are unsuitable, future research may require alternative techniques to overcome the robustness-accuracy tradeoff.

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

## A. Notations

We clarify the notations used in the paper.

- $H_\theta$: learned inverse mapping parameterized by $\theta$, i.e., black-box neural networks, MBAs. For MBAs in particular, $H_\theta = \arg\min_x \frac{1}{2}\|\boldsymbol{y} - \boldsymbol{Ax}\|_2^2 + r_\theta(\boldsymbol{x})$.

- $r$: regularization function from predefined functions, i.e., $\ell_1$ and $\ell_2$ norm.

- $r_\theta$: regularization function whose gradient is learned from a neural network $f_\theta$.

- $f_\theta$: learned gradient update of $r$, $f_\theta(\boldsymbol{x}) = \nabla_{\boldsymbol{x}} r_\theta(\boldsymbol{x})$.

- $F(x) = \frac{1}{2}\|\boldsymbol{y} - \boldsymbol{Ax}\|_2^2 + r_\theta(\boldsymbol{x})$, denote the value of the lower-level objective function.

## B. Proof of SGD Jittering Convergence (Corollary 6.2)

**Corollary B.1.** *(Restate Corollary 6.2. Convergence of MBAs-SGD for IPs) Let $F(\boldsymbol{x}) = \frac{1}{2}\|\boldsymbol{y} - \boldsymbol{Ax}\|_2^2 + r_\theta(\boldsymbol{x})$ denote the lower-level objective function. Assume $f_\theta = \nabla r_\theta$ is $L$-Lipschitz continuous, satisfying*

$$\|f_\theta(\boldsymbol{x} + \Delta) - f_\theta(\boldsymbol{x})\|_2 \leq L\|\Delta\|_2, \quad \forall \boldsymbol{x}, \Delta,$$

*let $\mu = \max_{i=1,..,n} \lambda_i$, where $\lambda_i$ are the eigenvalues of $\boldsymbol{A}^\top \boldsymbol{A}$, and define $L_{max} = \max\{\mu, L\}$. Consider a sequence $\{\boldsymbol{x}_k^{sgd}\}_{k=0}^K$ generated by SGD in (8) with a constant step-size $\eta = \sqrt{\frac{2}{LL_{max}K}}$. For $K \geq 1$, the following holds:*

$$\min_{0 \leq k < K} \mathbb{E}\|\nabla F(\boldsymbol{x}_k^{sgd})\|_2^2 \leq \sqrt{\frac{2LL_{max}}{K}} \left(2(F(\boldsymbol{x}_0^{sgd}) - \inf F) + \Delta_F^*\right),$$

*where, $\boldsymbol{x}^* = \arg\min_x F$, and*

$$\Delta_F^* = F(\boldsymbol{x}^*) - \inf\left(\frac{1}{2}\|\boldsymbol{y} - \boldsymbol{Ax}\|_2^2\right) - \inf r_\theta(\boldsymbol{x}).$$

*Proof.* The corollary follows directly from Theorem 5.12 in (Garrigos & Gower, 2023), which assumes the objective function is Lipschitz. In the case of the SGD jittering training scheme, the lower-level objective function is given by $F(\boldsymbol{x}) = \frac{1}{2}\|\boldsymbol{y} - \boldsymbol{Ax}\|_2^2 + r_\theta(\boldsymbol{x})$. We show that $F$ is $(L + \mu)$-Lipschitz, allowing the result from Theorem 5.12 to be directly applied.

Assume $f_\theta$ is $L$-Lipschitz, expanding the gradient of $F$, we have,

$$\|\nabla_{\boldsymbol{x}} F(\boldsymbol{x} + \Delta) - \nabla_{\boldsymbol{x}} F(\boldsymbol{x})\|_2$$
$$= \|\boldsymbol{A}^\top(\boldsymbol{A}(\boldsymbol{x} + \Delta) - \boldsymbol{y}) + f_\theta(\boldsymbol{x} + \Delta) - \boldsymbol{A}^\top(\boldsymbol{Ax} - \boldsymbol{y}) - f_\theta(\boldsymbol{x})\|_2$$
$$= \|\boldsymbol{A}^\top \boldsymbol{A}\Delta + f_\theta(\boldsymbol{x} + \Delta) - f_\theta(\boldsymbol{x})\|_2$$
$$\leq \|\boldsymbol{A}^\top \boldsymbol{A}\| \|\Delta\|_2 + \|f_\theta(\boldsymbol{x} + \Delta) - f_\theta(\boldsymbol{x})\|_2$$
$$\leq \max_{i=1,..,n} \lambda_i \|\Delta\|_2 + \|f_\theta(\boldsymbol{x} + \Delta) - f_\theta(\boldsymbol{x})\|_2,$$

where $\lambda_i$ be the eigenvalues of $\boldsymbol{A}^\top \boldsymbol{A}$, and let $\mu = \max_{i=1,..,n} \lambda_i$. Thus $F$ is $(L + \mu)$-Lipshitz and we conclude the proof.

$\square$

## C. Proof of Generalization (Theorem 7.4)

**Theorem C.1.** *(Restate Theorem 7.4) Assuming a small zero-meam Gaussian random vector $\boldsymbol{g}$ with $\mathbb{E}\|\boldsymbol{g}\|_2^2 = \sigma_g^2$ is added to data $\boldsymbol{x} \in \mathcal{D}$ during evaluation. The generalization risk for denoising problem is defined as,*

$$\mathcal{G}(\theta) = \mathbb{E}_{\boldsymbol{x}, \boldsymbol{y} \sim \mathcal{D}, \boldsymbol{g}} \|\boldsymbol{x} + \boldsymbol{g} - H_\theta(\boldsymbol{y} + \boldsymbol{g})\|_2^2. \tag{13}$$

*Under assumptions 6.1, 7.1 and 7.2, SGD jittering generalizes better than regular MSE training,*

$$\mathcal{G}(\theta^{sgd}) \leq \mathcal{G}(\theta^{mse}).$$

*Proof.* For any trained model $\theta$, we have generalization risk in (13). We can express $H_\theta(\boldsymbol{y} + \boldsymbol{g})$ in terms of $H_\theta(\boldsymbol{y})$ for the same set of parameters via iterative expansions. For iterations $k = 0, 1, ..., K$, let the sequence $\{\boldsymbol{x}_0, \boldsymbol{x}_1, ..., \boldsymbol{x}_K\}$ denote the denoising trajectory from observation $\boldsymbol{y} + \boldsymbol{g}$, where $H_\theta(\boldsymbol{y} + \boldsymbol{g}) = \boldsymbol{x}_K$, and let $\{\boldsymbol{x}'_0, \boldsymbol{x}'_1, ..., \boldsymbol{x}'_K\}$ denote the denoising trajectory from observation $\boldsymbol{y}$ where $H_\theta(\boldsymbol{y}) = \boldsymbol{x}'_K$. Then we can rewrite the reconstruction trajectory with the appearance of $\boldsymbol{g}$ as follows,

$$
\begin{aligned}
\boldsymbol{x}_0 &= \boldsymbol{x}'_0 + \boldsymbol{g} = \boldsymbol{y} + \boldsymbol{g} \\
\boldsymbol{x}_1 &= (1 - \eta)\boldsymbol{x}_0 + \eta(\boldsymbol{y} + \boldsymbol{g}) - \eta f_\theta(\boldsymbol{x}_0) \\
&= \big((1 - \eta)\boldsymbol{x}'_0 + \eta\boldsymbol{y} - \eta f_\theta(\boldsymbol{x}'_0)\big) + \big(\boldsymbol{g} + \eta f_\theta(\boldsymbol{x}'_0) - \eta f_\theta(\boldsymbol{x}_0)\big) \\
&= \boldsymbol{x}'_1 + \boldsymbol{g} + \eta(f_\theta(\boldsymbol{x}'_0) - f_\theta(\boldsymbol{x}_0)) \\
\boldsymbol{x}_2 &= (1 - \eta)\boldsymbol{x}_1 + \eta(\boldsymbol{y} + \boldsymbol{g}) - \eta f_\theta(\boldsymbol{x}_1) \\
&= \boldsymbol{x}'_2 + \boldsymbol{g} + \eta(1 - \eta)(f_\theta(\boldsymbol{x}'_0) - f_\theta(\boldsymbol{x}_0)) + \eta(f_\theta(\boldsymbol{x}'_1) - f_\theta(\boldsymbol{x}_1)) \\
&\qquad ... \\
\boldsymbol{x}_{k+1} &= \boldsymbol{x}'_{k+1} + \boldsymbol{g} + \sum_{i=0}^{k} \eta(1 - \eta)^{k-i}(f_\theta(\boldsymbol{x}'_i) - f_\theta(\boldsymbol{x}_i))
\end{aligned}
$$

Let $K = k + 1$ and plug in the above expansion to (9). For brevity, we omit the subscripts of expectations in the proof.

$$
\begin{aligned}
\mathcal{G}(\theta) &= \mathbb{E}\|\boldsymbol{x} - \boldsymbol{x}'_K - \sum_{i=0}^{K-1} \eta(1 - \eta)^{K-1-i}(f_\theta(\boldsymbol{x}'_i) - f_\theta(\boldsymbol{x}_i))\|_2^2 \\
&= \mathbb{E}\|\boldsymbol{x} - \boldsymbol{x}'_K\|_2^2 + \mathbb{E}\|\sum_{i=0}^{K-1} \eta(1 - \eta)^{K-1-i}(f_\theta(\boldsymbol{x}'_i) - f_\theta(\boldsymbol{x}_i))\|_2^2 - 2\mathbb{E}\big[\langle \boldsymbol{x} - \boldsymbol{x}'_K, \sum_{i=0}^{K-1} \eta(1 - \eta)^{K-1-i}(f_\theta(\boldsymbol{x}'_i) - f_\theta(\boldsymbol{x}_i))\rangle\big]
\end{aligned}
$$

The first term is the testing accuracy when $\boldsymbol{x}, \boldsymbol{y}$ are in the data distribution, while the second and third terms are affected by $\boldsymbol{g}$. MSE training minimizes the first term solely, but no extra regularization on the other terms. On the other hand, we will show that SGD jittering training also penalizes the magnitude of the other terms, thus obtaining a smaller generalization risk than MSE training.

We view the SGD jittering process as a noisy version of regular MSE training, the goal is to write the SGD jittering risk in terms of the noiseless updates with some implicit regularization. Let $\{\boldsymbol{x}'_0, \boldsymbol{x}'_1, ..., \boldsymbol{x}'_K\}$ denote the noiseless GD trajectory of a model $\theta$. We write the iterative updates of SGD jittering training $\{\boldsymbol{x}_0^{sgd}, \boldsymbol{x}_1^{sgd}, ..., \boldsymbol{x}_K^{sgd}\}$ for the same set of parameters $\theta$ in terms of $\boldsymbol{x}'_k$s, which tells the deviation of each intermediate reconstruction between the SGD jittering and its noiseless counterpart,

$$
\begin{aligned}
\boldsymbol{x}_0^{sgd} &= \boldsymbol{y} = \boldsymbol{x}'_0 + \boldsymbol{w}_0 \\
\boldsymbol{x}_1^{sgd} &= (1 - \eta)\boldsymbol{x}_0^{sgd} + \eta\boldsymbol{y} - \eta f_\theta(\boldsymbol{x}_0^{sgd}) - \eta\boldsymbol{w}_1 \\
&= \boldsymbol{x}'_1 + \eta(f_\theta(\boldsymbol{x}'_0) - f_\theta(\boldsymbol{x}_0^{sgd})) - (\eta(1 - \eta)\boldsymbol{w}_0 + \eta\boldsymbol{w}_1) \\
\boldsymbol{x}_2^{sgd} &= (1 - \eta)\boldsymbol{x}_1^{sgd} + \eta\boldsymbol{y} - \eta f_\theta(\boldsymbol{x}_1^{sgd}) - \eta\boldsymbol{w}_2 \\
&= \boldsymbol{x}'_2 + \eta\big(f_\theta(\boldsymbol{x}'_1) - f_\theta(\boldsymbol{x}_1^{sgd})\big) + \eta(1 - \eta)\big(f_\theta(\boldsymbol{x}'_0) - f_\theta(\boldsymbol{x}_0^{sgd})\big) - \big(\eta(1 - \eta)^2\boldsymbol{w}_0 + \eta(1 - \eta)\boldsymbol{w}_1 + \eta\boldsymbol{w}_2\big) \\
&\qquad ... \\
\boldsymbol{x}_{k+1}^{sgd} &= (1 - \eta)\boldsymbol{x}_k^{sgd} + \eta\boldsymbol{y} - \eta f_\theta(\boldsymbol{x}_k^{sgd}) - \eta\boldsymbol{w}_{k+1} \\
&= \boldsymbol{x}'_{k+1} + \sum_{i=0}^{k} \eta(1 - \eta)^{k-i}(f_\theta(\boldsymbol{x}'_i) - f_\theta(\boldsymbol{x}_i^{sgd})) - \sum_{i=0}^{k+1} \eta(1 - \eta)^{k+1-i}\boldsymbol{w}_i.
\end{aligned}
$$

Let $K = k + 1$. Since the jittering noise is zero-mean, the SGD jittering training risk becomes,

$$
\begin{aligned}
\mathbb{E}\|\boldsymbol{x} - \boldsymbol{x}_K^{sgd}\|_2^2 &= \mathbb{E}\|\boldsymbol{x} - \boldsymbol{x}'_K\|_2^2 + \mathbb{E}\|\sum_{i=0}^{K} \eta(1 - \eta)^{K-i}\boldsymbol{w}_i\|_2^2 + \mathbb{E}\|\sum_{i=0}^{K-1} \eta(1 - \eta)^{K-1-i}(f_\theta(\boldsymbol{x}'_i) - f_\theta(\boldsymbol{x}_i^{sgd}))\|_2^2 \\
&\quad - 2\mathbb{E}\left[\langle \boldsymbol{x} - \boldsymbol{x}'_K, \sum_{i=0}^{K-1} \eta(1 - \eta)^{K-1-i}(f_\theta(\boldsymbol{x}'_i) - f_\theta(\boldsymbol{x}_i^{sgd}))\rangle\right].
\end{aligned}
$$

The first component corresponds to the MSE loss when the trajectory is not perturbed by jittering noise. The second term is weighted noise variance independent of the network parameters. The third term adds extra regularization to the difference in

outputs of $f_\theta$ between noisy and clean trajectories. It penalizes more as the iterations approach the final output. The last term is the cross product between the reconstruction error under noiseless GD trajectory and the perturbed reconstructions from $f$, which is lower bounded since the overall SGD jittering risk is non-negative.

When choosing $\sigma_{\boldsymbol{w}_k}^2$ at iteration $k$ such that $\sum_{i=0}^{k} \eta^2(1-\eta)^{2(k-i)}\sigma_{w_i}^2 = \sigma_{\boldsymbol{g}}^2$, the reconstruction from $\boldsymbol{x}+\boldsymbol{g}$ is a special case in SGD jittering training where $\boldsymbol{w}_k$ are independently sampled at each iteration. Thus, minimizing the regularization term $\mathbb{E}_{(\boldsymbol{x},\boldsymbol{y})\sim\mathcal{D},\boldsymbol{w}_k\forall k} \,||\sum_{i=0}^{K-1}\eta(1-\eta)^{K-1-i}(f_\theta(\boldsymbol{x}_i') - f_\theta(\boldsymbol{x}_i^{sgd}))||_2^2$ also reduces the penalty term in computing generalization accuracy $\mathbb{E}_{\boldsymbol{x},\boldsymbol{y}\in\mathcal{D},\boldsymbol{g}}||\sum_{i=0}^{K-1}\eta(1-\eta)^{K-1-i}(f_\theta(\boldsymbol{x}_i') - f_\theta(\boldsymbol{x}_i))||_2^2$. Minimizing the last term also minimizes $-2\mathbb{E}\big[\langle \boldsymbol{x} - \boldsymbol{x}_K', \sum_{i=0}^{K-1}\eta(1-\eta)^{K-1-i}(f_\theta(\boldsymbol{x}_i') - f_\theta(\boldsymbol{x}_i))\rangle$ in the generalization accuracy. Therefore, $\mathcal{G}(\theta^{sgd}) \leq \mathcal{G}(\theta^{mse})$ due to the implicit regularization term in SGD jittering training.

$\square$

## D. Proof of Robustness (Theorem 7.5)

**Theorem D.1.** *(Restate Theorem 7.5) Let $\boldsymbol{e}$ be the perturbation vector and let $\mathcal{P}_e$ be iid zero-mean Gaussian distribution such that $\mathbb{E}||\boldsymbol{e}||_2^2 = \sigma_e^2$. The average-case robustness risk for denoising problem is as follows,*

$$R_e(\theta) = \mathbb{E}_{(\boldsymbol{x},\boldsymbol{y})\sim\mathcal{D},\boldsymbol{e}\sim\mathcal{P}_e} \left[ ||\boldsymbol{x} - H_\theta(\boldsymbol{y}+\boldsymbol{e})||_2^2 \right]. \tag{14}$$

*For the denoising problem, SGD jittering is more robust than MSE training against bounded-variance perturbations around the measurement, or*

$$R_e(\theta^{sgd}) \leq R_e(\theta^{mse}).$$

*Proof.* The proof is similar to proving generalization accuracy. To evaluate the average-case robustness for any trained $\theta$, we first write the iterative updates under attack $\{\boldsymbol{x}_0^{at}, \boldsymbol{x}_1^{at}, ...\boldsymbol{x}_K^{at}\}$ in terms of the noiseless trajectory $\{\boldsymbol{x}_0', \boldsymbol{x}_1', ...\boldsymbol{x}_K'\}$.

$$\begin{aligned}
\boldsymbol{x}_0^{at} &= \boldsymbol{x}_0' + \boldsymbol{e} = \boldsymbol{y} + \boldsymbol{e} \\
\boldsymbol{x}_1^{at} &= (1-\eta)\boldsymbol{x}_0^{at} + \eta(\boldsymbol{y}+\boldsymbol{e}) - \eta f_\theta(\boldsymbol{x}_0^{at}) \\
&= \boldsymbol{x}_1' + \boldsymbol{e} + \eta(f_\theta(\boldsymbol{x}_0') - f_\theta(\boldsymbol{x}_0^{at})) \\
\boldsymbol{x}_2^{at} &= (1-\eta)\boldsymbol{x}_1^{at} + \eta(\boldsymbol{y}+\boldsymbol{e}) - \eta f_\theta(\boldsymbol{x}_1^{at}) \\
&= \boldsymbol{x}_2' + \boldsymbol{e} + \eta(1-\eta)(f_\theta(\boldsymbol{x}_0') - f_\theta(\boldsymbol{x}_0^{at})) + \eta(f_\theta(\boldsymbol{x}_1') - f_\theta(\boldsymbol{x}_1^{at})) \\
&\quad ... \\
\boldsymbol{x}_{k+1}^{at} &= \boldsymbol{x}_{k+1}' + \boldsymbol{e} + \sum_{i=0}^{k} \eta(1-\eta)^{k-i}(f_\theta(\boldsymbol{x}_i') - f_\theta(\boldsymbol{x}_i^{at}))
\end{aligned}$$

Let $K = k+1$, we rewrite the robustness risk. Notice that the generalization loss in (9) includes the additional vector $\boldsymbol{g}$, whereas the robustness risk in (11) does not.

$$\begin{aligned}
R_e(\theta) &= \mathbb{E}||\boldsymbol{x} - \boldsymbol{x}_K' - \boldsymbol{e} - \sum_{i=0}^{K-1}\eta(1-\eta)^{K-1-i}(f_\theta(\boldsymbol{x}_i') - f_\theta(\boldsymbol{x}_i^{at}))||_2^2 \\
&= \mathbb{E}||\boldsymbol{x} - \boldsymbol{x}_K'||_2^2 + \mathbb{E}||\sum_{i=0}^{K-1}\eta(1-\eta)^{K-1-i}(f_\theta(\boldsymbol{x}_i') - f_\theta(\boldsymbol{x}_i^{at}))||_2^2 + \mathbb{E}||\boldsymbol{e}||_2^2 - 2\mathbb{E}\left[\langle \boldsymbol{x} - \boldsymbol{x}_K', \sum_{i=0}^{K-1}\eta(1-\eta)^{K-1-i}(f_\theta(\boldsymbol{x}_i') - f_\theta(\boldsymbol{x}_i^{at}))\rangle\right] \\
&= \mathbb{E}||\boldsymbol{x} - \boldsymbol{x}_K'||_2^2 + \mathbb{E}||\sum_{i=0}^{K-1}\eta(1-\eta)^{K-1-i}(f_\theta(\boldsymbol{x}_i') - f_\theta(\boldsymbol{x}_i^{at}))||_2^2 + \sigma_e^2 - 2\mathbb{E}\left[\langle \boldsymbol{x} - \boldsymbol{x}_K', \sum_{i=0}^{K-1}\eta(1-\eta)^{K-1-i}(f_\theta(\boldsymbol{x}_i') - f_\theta(\boldsymbol{x}_i^{at}))\rangle\right]
\end{aligned}$$

The first term is the mean-squared testing loss for in-distribution data. The second term is an additional penalty that appears in SGD jittering training. When picking $\sigma_{\boldsymbol{w}_k}^2$ for all $k$ and $\sigma_e^2$ such that $\sum_{k=0}^{k}\eta^2(1-\eta)^{2(k-i)}\sigma_{w_i}^2 = \sigma_e^2$, this term in robustness risk is a special case in SGD jittering risk. The last term also appears in the SGD jittering risk; therefore, minimizing the SGD jittering risk implicitly reduces this term as well. Therefore, for models that perform equally well in MSE for in-distribution testing data, minimizing SGD jittering risk improves the robustness. $\square$

# E. Extra Stochastic Proximal Gradient Descent Experiments

Similar to gradient descent variants of loop unrolling architectures, for proximal gradient descent (PGD) variants, we train a PGD LU with regular MSE loss. Then finding the worst-case attack to $y$ over all PGD updates.

## E.1. Seismic Deconvolution

In the same seismic deconvolution problem setup discussed in the main manuscript, but now solved using proximal gradient variants, Figure 7(a) shows the performance of the trained models under an adversarial attack with $\epsilon = 1$. Figure 7(b) focuses on reconstructing the additional layer highlighted in red boxes. Table 3 provided numerical evaluations. The proposed SPGD jittering demonstrates greater robustness compared to MSE training while better preserving detailed information than adversarial training (AT).

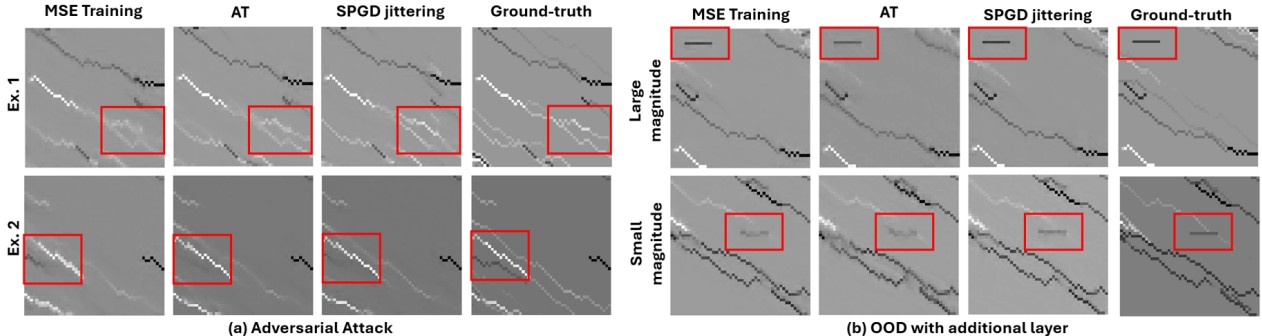

*Figure 7.* Visualization of seismic deconvolution using (stochastic) **proximal gradient descent** variants within loop unrolling architectures. Figure (a) illustrates the reconstruction performance under adversarial attack with $\epsilon = 1$. Figure (b) evaluates the model's performance on out-of-distribution (OOD) seismic data, where the ground-truth reflectivity includes an additional small horizontal layer, highlighted in the red boxes.

| PSNR / SSIM | in-distribution | Adv. Attack | OOD |
|---|---|---|---|
| MSE training | 34.91 / 0.923 | 28.78 / 0.842 | 34.810 / 0.920 |
| AT | 32.99 / 0.899 | **31.44** / **0.875** | 32.966 / 0.899 |
| Input Jittering | 33.02 / 0.899 | 30.56 / 0.861 | 32.910 / 0.895 |
| SPGD Jittering | **35.05** / **0.931** | 29.93 / 0.870 | **34.952** / **0.929** |

*Table 3.* PGD variants of LU are evaluated on seismic deconvolution across three scenarios: in-distribution data (column 2), adversarial attacks (column 3), and out-of-distribution (OOD) data (column 4). The best performance in each case is highlighted in bold, while the second-best performance is underlined.

## E.2. Accelerated MRI Reconstruction

We complete the missing numerical results discussed in the main manuscript for $4\times$ accelerated MRI reconstruction using SPGD jittering in Table 4.

| PSNR / SSIM | fastMRI | Adv. Attack | Tumor Cell |
|---|---|---|---|
| MSE training | 27.93 / 0.582 | 22.82 / 0.179 | 29.39 / 0.771 |
| AT | 27.20 / 0.533 | 26.78 / 0.518 | 27.13 / 0.560 |
| Input Jittering | 24.88 / 0.299 | 24.44 / 0.297 | 25.85 / 0.478 |
| SPGD Jittering | **28.15** / **0.597** | **27.56** / **0.577** | **29.762** / **0.777** |

*Table 4.* PGD variants of LU are evaluated on MRI data across three scenarios: in-distribution data (column 2), adversarial attacks (column 3), and out-of-distribution (OOD) data (column 4). The best performance in each case is highlighted in bold, while the second-best performance is underlined.

## F. Compared to Other Baselines for MRI reconstruction

We evaluate our method against a widely used robustness technique Lipschitz regularization using spectral normalization (SN). SN constrains the Lipschitz constant of the network by bounding the spectral norm (i.e., largest singular value) of each weight matrix, which helps stabilize training and improve robustness. As shown in the second last row in Table 5 for 4xMRI reconstruction. SN promotes adv. robustness, but at the cost of reduced accuracy to some extend. This is likely due to its restrictive nature of SN limiting the model's expressive power.

Additionally, we also compares the performance to a diffusion models (DMs), as they also demonstrates impressive results in image generation. Since our work proposes a general framework for IPs, we chose to compare against the standard DDPM rather than specialized task-specific DMs, consistent with prior work (Güngör et al., 2023). To ensure a fair comparison under similar computational constraints, we adapt the denoising U-Net to fit within the same GPU memory as other methods.

Table 5 compares DDPM with other methods. DDPM achieves comparable in-distribution (ID) performance to MBAs trained with both MSE loss and the proposed SGD jittering, and shows stronger robustness than standard MSE-trained MBAs. However, it underperforms in OOD generalization. While DDPM serves as a strong baseline for robustness and ID accuracy, SGD jittering achieves better generalization under distribution shifts. We will add the comparison to DM in the main manuscript as an interesting baseline.

It is also worth noting that DDPM requires 10× more parameters and is significantly more data-intensive, whereas MBAs are more data-efficient (Monga et al., 2021) due to their optimization-inspired iterative structure. We also refer to prior work (Güngör et al., 2023), which compares DDPM, DiffRecon, AdaDiff to MSE-trained MBAs for MRI reconstruction. Their results show that while DM can generalize well in some cases, MBA methods consistently perform better on ID data. Results in (Güngör et al., 2023) shows that MSE-trained MBA is a strong baseline, and our proposed SGD jittering further improves their robustness and generalization.

| Model | Training Scheme/Design | fastMRI | Adv. Attack | Tumor Cell |
|-------|------------------------|---------|-------------|------------|
| GD LU | MSE training | 28.21 / 0.603 | 25.68 / 0.382 | 29.92 / 0.779 |
| | AT | 27.68 / 0.564 | **27.17** / 0.549 | 27.74 / 0.597 |
| | Input Jittering | 28.18 / 0.595 | 25.05 / 0.420 | 29.97 / 0.740 |
| | SGD jittering (Ours) | **28.22** / 0.607 | 26.77 / **0.552** | **30.36** / **0.788** |
| | Lipschitz regularization (SN) | 27.71 / 0.576 | 27.09 / 0.542 | 27.92 / 0.594 |
| DDPM | Default training | 28.17 / **0.611** | 27.13 / 0.536 | 29.72 / 0.782 |

*Table 5.* MRI reconstruction evaluation compared to the Lipschitz regularization method (last row). The best and second best performances are in bold and underlined respectively.

## G. Training Details

We use 10-iteration gradient descent loop unrolling architectures for all tasks. For the toy example, we use a 3-layer MLP with a hidden dimension of 32 as the learned gradient network. For seismic deconvolution and MRI reconstruction, a 5-layer and 8-layer DnCNN with 64 hidden channels are used for the learned gradient network, respectively. All models are trained using Nvidia RTX3080, using Adam optimizer with a learning rate of $1e-4$.

The jittering noise variance for each method is selected based on performances of both robustness and accuracy. In input jittering training, the three methods in order of 2D denoising, seismic deconvolution and MRI reconstruction use the noise variance of 0.01, 0.05 and 0.05. In SGD jittering training, we choose the SGD jittering variance for each task with 0.01, 0.1 and 0.01 respectively.

## H. Training Speed

We also compare the training speed among different training schemes for GD and PGD LU architectures. All methods are trained on the same device. Training batch sizes for the 2D denoising problem, seismic deconvolution and MRI are 256, 16, and 4 respectively. AT is trained by projected gradient descent. For the 2D denoising problem, $\epsilon = 0.01$ with 50 projected GD iterations and a stepsize of 0.05. For the seismic deconvolution and MRI reconstruction problem, $\epsilon = 1$, with 20 projected GD iterations and a stepsize of 0.1. AT is significantly slower than other methods. Notice that the jittering in SGD and SPGD does introduce very minimal time overhead compared to standard MSE training, even when accounting for

the additional noise sampling steps.

|        |                 | 2D denoising   | Seis. Deconv. | MRI        |
|--------|-----------------|----------------|---------------|------------|
| GD LU  | MSE training    | 5360.84 it/s   | 23.69 it/s    | 5.23 it/s  |
|        | AT              | 562.75 it/s    | 1.53 it/s     | 0.25 it/s  |
|        | Input Jittering | 5256.77 it/s   | 22.80 it/s    | 5.20 it/s  |
|        | SGD jittering   | 4337.84 it/s   | 22.72 it/s    | 5.27 it/s  |
| PGD LU | MSE training    | 5136.34 it/s   | 16.78 it/s    | 6.23 it/s  |
|        | AT              | 486.68 it/s    | 1.12 it/s     | 0.17 it/s  |
|        | Input Jittering | 4965.25 it/s   | 16.44 it/s    | 6.17 it/s  |
|        | SPGD jittering  | 4669.76 it/s   | 14.52 it/s    | 6.07 it/s  |

*Table 6.* Training speed measured in items per second are recorded.

