# OpenReview forum: "SGD Jittering: A Training Strategy for Robust and Accurate Model-Based Architectures"
_ICML.cc/2025/Conference — ICML 2025 poster_

### Official Review · Reviewer_x3Tj · 2025-03-16

**Overall Recommendation:** 3

**Summary:**

The paper introduces SGD Jittering, a training method for model-based architectures (MBAs) solving inverse problems. By adding small, random noise to gradient updates during training, SGD Jittering improves robustness and generalization accuracy without modifying input data or increasing computational cost like adversarial training (AT). Theoretical analysis proves its advantages over standard mean squared error (MSE) training. Experiments on denoising, seismic deconvolution, and MRI reconstruction show superior performance.

**Claims And Evidence:**

The claims are supported by clear evidence and proofs.

**Essential References Not Discussed:**

The key contribution is similar to stochastic gradient Langevin dynamics (SGLD), the the comparison between the proposed method and SGLD is not discussed.

**Experimental Designs Or Analyses:**

Experiments are ok but can be improved when more inverse tasks like superresolution are added.

**Methods And Evaluation Criteria:**

The method and evaluations make sense, but the method is similar to previous works like SGLD.  Therefore, the novelty is not very good.

**Other Comments Or Suggestions:**

The section structure should be reorganized. e.g., Sec. 3 only has one paragraph.

**Other Strengths And Weaknesses:**

Strengths:
1. Provides analysis linking noise injection to implicit regularization of gradient/Hessian smoothness, advancing understanding of robustness in iterative inverse solvers.
2. Demonstrates effectiveness on real-world tasks (seismic deconvolution, MRI) where robustness and generalization are critical. The extension to proximal gradient (SPGD Jittering) highlights broader applicability.
3. Well-structured presentation of theory, experiments, and ablation studies.

Weaknesses:

1. SGD jittering is similar to stochastic gradient Langevin dynamics, which makes the method less novel. In addition, the theoretical or empirical differences are not discussed.
2. Theoretical guarantees are limited to denoising tasks; broader analysis for general inverse problems (e.g., ill-posedness, nonlinearity) is not provided.
3. Experiments focus on a few inverse problems. Testing on more diverse tasks (e.g., super-resolution) could better validate generality.
4. While outperforming MSE and adversarial training, comparisons to other robustness methods (e.g., gradient penalty, Lipschitz regularization) are missing.

**Questions For Authors:**

Please see the weakness part.

**Relation To Broader Scientific Literature:**

The key contributions are similar to Stochastic gradient Langevin dynamics.

**Theoretical Claims:**

I checked some proofs but not all proofs.

---

> ### Author Rebuttal · Authors · 2025-03-31
>
> We thank the reviewer for their thoughtful comments and helpful questions. Please find detailed responses below.
>
> > Comparing to SGLD
>
> We thank R-x3Tj for mentioning SGLD, but we clarify that our SGD jittering is fundamentally different from SGLD in both goal and mechanisms.
>
> SGLD adds noise directly to network parameter updates to approximate sampling from the Bayesian posterior. It aims to capture model uncertainty, not necessarily to improve robustness or generalization. This process requires careful scheduling of the noise and often results in slower training.
>
> In contrast, SGD jittering injects noise into the hidden inputs (intermediate activations) during training—not into the parameters—with the explicit goal of enhancing robustness and generalization in IPs. Importantly, no noise is used at inference time, so reconstructions remain deterministic. Moreover, our method integrates naturally with MBAs and does not require any modification to the optimizer.
>
> > Theoretical guarantees are limited to denoising tasks; provide broader analysis for general IPs
>
> We agree that extending the theoretical guarantees to more general IPs, including nonlinear and ill-posed settings with data/model mismatch, remains a rich direction for future work. We see our current analysis as an important first step toward building a theoretical foundation for robust training schemes in MBAs, and we hope it will inspire further progress in this area.
>
> > Testing on more diverse tasks (e.g., super-resolution) could better validate generality
>
> In addition to the main experiments, we also evaluated our method on a natural image deblurring task using the CelebA dataset. Due to space constraints, these results were not included in the main text. Below, we provide test performance on in-distribution (CelebA), out-of-distribution (FairFace), and adversarial settings. Table 2 shows improved ID/OOD accuracies and robustness over MSE training, with robustness nearly matching AT. We will include these results and discussion in the revised manuscript to further support the generality of our approach.
>
> | PSNR/SSIM            | ID-CelebA         | Adv. Attack       | OOD-FairFace      |
> |----------------------|-------------------|-------------------|-------------------|
> |     MSE training     |   <34.14 / 0.954>   |   29.81 / 0.812   |   <32.94 / 0.940>   |
> |          AT          |   32.10 / 0.928   | **31.83 / 0.902** |   31.26 / 0.918   |
> |    Input Jittering   |   34.06 / 0.942   |   31.28 / 0.857   |   31.04 / 0.912   |
> | SGD jittering (Ours) | **35.12 / 0.960** |   <31.46 / 0.884>   | **33.23 / 0.945** |
> Table 2: Image deblurring. Best performances in **bold**, second best in <...>.
>
> > Other robustness methods
>
> Thank the reviewer for the suggestion. Prior work [1] has compared a MBAs trained with AT to end-to-end randomized smoothing (RS) for MRI reconstruction, and showed that AT improves robustness significantly than RS. It supports our choice of **AT as a strong baseline for evaluating robustness in IPs**.
>
> As suggested, we evaluated our method against a widely used robustness technique Lipschitz regularization using spectral normalization (SN). SN constrains the Lipschitz constant of the network by bounding the spectral norm (i.e., largest singular value) of each weight matrix, which helps stabilize training and improve robustness. As shown in Table 3 result for 4xMRI reconstruction. SN promotes adv. robustness, but at the cost of reduced accuracy to some extend. This is likely due to its restrictive nature of SN limiting the model’s expressive power.
>
> |    PSNR/SSIM     | ID | Adv. Attack | OOD |
> |:--------------------:|:-------------:|:-------------:|:-------------:|
> |     MSE training     | <28.21 / 0.603> | 25.68 / 0.382 | 29.92 / <0.779> |
> |          AT          | 27.68 / 0.564 | **27.17** / <0.549> |  27.74 / 0.597 |
> |    Input Jittering   | 28.18 / 0.595 | 25.05 / 0.420 | <29.97> / 0.740 |
> | SGD jittering (Ours) | **28.22 / 0.607** | 26.77 / **0.552** | **30.36 / 0.788** |
> |  SN (new baseline) | 27.71 / 0.576  | <27.09> / 0.542 | 27.92 / 0.594 |
> Table 3: MRI reconstruction. Best performances in **bold**, second best in <...>.
>
> [1] Alkhouri et al. Robust physics-based deep MRI reconstruction via diffusion purification
>
> We sincerely appreciate the reviewer’s constructive feedback. We believe the added experiments and clarification will strengthened the manuscript and hope they satisfactorily address your concerns.

---

### Official Review · Reviewer_mUA8 · 2025-03-21

**Overall Recommendation:** 3

**Summary:**

The paper introduces "SGD Jittering," a new  training strategy designed to enhance the robustness and generalization of Model-Based Architectures (MBAs) for image inverse problems. Specifically, the authors propose to inject random zero-mean Gaussian noises into gradient updates at each iteration within deep unrolling networks during training. Theoretically, they demonstrate that this simple noise injection improves average-case robustness and generalization accuracy compared to standard mean-squared-error and adversarial training, respectively. Empirically, they validate their method across several inverse problem tasks, including a toy denoising example, seismic deconvolution, and single-coil MRI reconstruction.

**Claims And Evidence:**

Yes. The paper provides both empirical and theoretical results on non-convex deep neural networks, systematically comparing SGD Jittering with standard MSE training and worst-case adversarial attacks. Additionally, to the best of the reviewer's knowledge, this work is the first to offer a theoretical analysis of generalization accuracy in inverse problems, particularly in the presence of small perturbations in test data.

**Essential References Not Discussed:**

N/A

**Experimental Designs Or Analyses:**

Yes, the empirical design is methodologically sound and provides informative insights.

**Methods And Evaluation Criteria:**

Yes, particularly in the empirical evaluation, where the model is trained on the fastMRI dataset and tested on a different knee dataset from Bickle & Jin (2021). Note that the test set contains giant tumor cells absent from the training data, ensuring a robust out-of-distribution (OOD) evaluation.

**Other Comments Or Suggestions:**

N/A

**Other Strengths And Weaknesses:**

Strengths

1, This paper presents an interesting and easily implementable training strategy to improve robustness and generalization in model-based deep learning.

2, The paper is well-written and easy to follow, with clear mathematical formulations and consistent notations.

3, The theoretical analysis effectively supports the intuition behind the proposed method, explicitly linking SGD jittering to improved robustness and generalization.

Weaknesses

1, The main theorems, 7.4 and 7.5, are primarily derived for image denoising, leaving their applicability to more general inverse problems (e.g., MRI or seismic reconstruction) unclear. Extending the theory may require additional assumptions on the forward model.

2, The results on proximal-based MBAs appear to be an ad-hoc empirical extension, rather than a direct consequence of the theoretical analysis. It is unclear whether the current theoretical framework applies directly or if additional assumptions are needed for deep neural network priors.

3, The paper lacks a more comprehensive review of generalization and robustness in inverse problems. While Definition 3.2 introduces a generalization risk formulation, it does not fully account for forward model mismatches between training and inference, which are critical in practical applications.

**Questions For Authors:**

1, Can the authors clarify what regularization functionals are used for constructing $r_\theta$ ?

2, Can similar approach and theoretical analysis to deep priors without explicit potentials, such as those based on regularization by denoising (RED) frameworks ?

3, Can the authors further clarify why the chosen definition of generalization risk (Eq. (3)) is suitable for general inverse problems? Would other definitions (such as those involving a distributional shift of the forward operator $A$ or the noise $z$) be potentially more relevant?

**Relation To Broader Scientific Literature:**

Prior works, Fawzi et al., 2016, showed that small perturbations in data space can significantly degrade performance, particularly for deep networks trained on ill-posed inverse problems. This paper’s approach of introducing SGD perturbations at the optimization level offers a new perspective.

**Theoretical Claims:**

The assumptions used for the theorems align well with existing literature. The training convergence analysis under the SGD optimizer is a direct extension of Garrigos & Gower (2023). Additionally, I do not see major flaws in Theorems 7.2 and 7.4.

---

> ### Author Rebuttal · Authors · 2025-03-31
>
> We thank the reviewer for their thoughtful comments and helpful questions. Please find detailed responses below.
>
> > Generality of Theoretical Results
>
> We agree that Theorems 7.4 and 7.5 were established specifically in the denoising setting. Extending the theoretical analysis to more complex inverse problems would require additional assumptions and broader treatment of the forward model. We will explicitly acknowledge this limitation in the revised manuscript and outline it as a promising direction for future work.
>
> > Extension to Proximal-Based MBAs
>
> The analysis of robustness and generalization for SPGD are not a direct extension of the current SGD theoretical results.
> While GD-LU and PGD-LU differ structurally—based on whether the consistency update and neural network modules (which learns $\nabla r$ in GD-LU or acts as a proximal operator in PGD-LU) are applied in parallel or sequentially—both architectures allow for noise injection into hidden states during training. This perturbs intermediate representations, thus promoting robustness.
>
> While our theoretical analysis focuses on SGD, prior work has established convergence for both SGD and SPGD in related setting. These results ensure that the reconstructed outputs remain consistent with the forward model $A$, thus preserving reconstruction accuracy. Motivated by this, we extend our approach empirically to the proximal setting. Our experiments confirm that jittering improves both robustness and generalization in proximal settings.
>
> > Generalization Risk Definition Eq.3
>
> Our current definition aims at capturing small shifts in test data, but assuming same and known forward model is used at inference. We agree that explicitly addressing forward-model mismatches between training and inference would be an important direction. This work serves as an initial discussion in addressing generalization and robustness issue only due to data-shift.
>
> > Clarification of Regularization Functional $r$
>
> In model-based architectures, the regularization function $r$ is implicitly defined through its gradient learned by a neural network $NN_\theta = \nabla r$. Thus, we do not explicitly specify the functional form of $r$. Please refer to L:117 under Eq.4 for a detailed discussion.
>
> > Clarifying Connections to RED frameworks
>
> Their motivation and implementation are fundamentally different. MBA is a supervised learning strategy whose training is formulated as a bilevel optimization problem (Eq.5-Eq.8), where the network is trained end-to-end and allows for noise injection during training. In contrast, RED uses a pre-trained denoiser as an explicit regularization function during inference, without end-to-end training. The denoiser in RED is fixed and not learned as part of the reconstruction process, so SGD jittering is not applicable to RED.
>
> > Review of robustness, generalization for IPs
>
> Thank the reviewers suggestion, and we will include the following paragraph in the manuscript:
>
> "To improve robustness, several strategies have been proposed, including training-time exposure to diverse perturbations such as noise injection and data augmentation [1-2], and adversarial training, enabling models to better handle noise during inference. Other approaches focus on architectural innovations—for example, diffusion models have shown inherent robustness to noise in MRI reconstruction [4], while PINN [5] embed domain knowledge to enhance stability without compromising interpretability. To improve generalization in IPs, methods like data augmentation with synthetic perturbations [6] and domain adaptation via techniques such as CycleGAN [7] have been used to expand the training distribution and improve adaptability to unseen data. Additionally, incorporating geometric constraints, as in [8] for electrocardiographic image reconstruction, has shown improved generalization by embedding prior knowledge into the learning process. While these approaches typically target either robustness or generalization, achieving both simultaneously remains an open and challenging problem."
>
> [1] Krainovic et al. Learning Provably Robust Estimators for Inverse Problems via Jittering
>
> [2] Zhou et al. Towards Understanding the Importance of Noise in Training Neural Networks
>
> [4] Dar et al. Adaptive diffusion priors for accelerated MRI reconstruction
>
> [5] Peng et al. Robust Regression with Highly Corrupted Data via Physics Informed Neural Networks
>
> [6] Guan et al. Solving Inverse Problems with Model Mismatch using Untrained Neural Networks within Model-based Architectures
>
> [7] Zhu et al. Unpaired Image-to-Image Translation using Cycle-Consistent Adversarial Networks
>
> [8] Jiang et al. Improving Generalization by Learning Geometry-Dependent and Physics-Based Reconstruction of Image Sequences
>
> We sincerely appreciate the reviewer’s constructive feedback. We believe the added literature review and clarification will strengthened the manuscript and hope they satisfactorily address your concerns.

---

> > ### Comment · Reviewer_mUA8 · 2025-04-08
> >
> > I thank the authors for their rebuttal, and I’m maintaining my original rating.

---

### Official Review · Reviewer_v9Xr · 2025-03-22

**Overall Recommendation:** 3

**Summary:**

The authors study the robustness and generalization properties of model-based architectures. The goal is to solve inverse problems with interpretable algorithms, such as loop-unrolling networks, and maintain two desirable properties: i) robustness to adversarial attacks, ii) generalization to small natural shifts in test-time data. The authors propose an algorithm called SGD jittering that makes progress in achieving these properties without sacrificing performance. The algorithm is based on a noisy version of Gradient Descent in the loop-unrolling architecture.

**Claims And Evidence:**

The claims are supported by the evidence.

**Essential References Not Discussed:**

N/A.

**Experimental Designs Or Analyses:**

I checked the experimental results (seismic deconvolution and MRI). As I mentioned above, I believe the biggest weakness is the lack of stronger baselines.

**Methods And Evaluation Criteria:**

The authors only include comparisons to other model-based algorithms. I think it would be better if stronger baselines were included, such as solving these problems with diffusion models. The latter lack the interpretability of model-based methods, but it would be nice to see what price we pay in performance to gain this interpretability. It would also be nice to include examples of where this interpretability is important.

**Other Comments Or Suggestions:**

N/A.

**Other Strengths And Weaknesses:**

Some weaknesses are mentioned above. I would further add that it is unclear what is the fundamental limit on the trade-off between generalization and robustness. How should the reader think of $x_g$ in Section 3? As an adversarial input or as a natural shift of $x$?
It would also be nice for the paper to include some convincing evidence on why model-based algorithms are interpretable in the context of MRI/Seismic Deconvolution and compare the performance with stronger baselines.

In terms of strengths, the paper has interesting theoretical results, it is well-presented and it has a clear motivation.

**Questions For Authors:**

Could you include comparisons with stronger baselines for the problems you want to address? MBAs do not need to necessarily improve upon these baselines, but it needs to be more clear what's the trade-off between interpretability, performance, robustness and generalization.

**Relation To Broader Scientific Literature:**

The paper improves the robustness and generalization of model-based algorithms that offer interpretable solutions to inverse problems.

**Theoretical Claims:**

I did not rigorously check all the proofs, but I read the theoretical results and the assumptions and they make sense.

---

> ### Author Rebuttal · Authors · 2025-03-31
>
> We thank the reviewer for the insightful suggestion, and please find the point-to-point response below.
>
> > Stronger baselines such as diffusion models (DM)
>
> We agree that diffusion models (DMs) have demonstrated impressive results in image generation. In response, we added DDPM-based experiments for MRI reconstruction. To ensure a fair comparison under similar computational constraints, we adapted the denoising U-Net to fit within the same GPU memory as other methods. Since our work proposes a general framework for IPs, we chose to compare against the standard DDPM rather than specialized task-specific DMs, consistent with prior work [1].
>
> Table 1 compares DDPM with other methods. DDPM achieves comparable in-distribution (ID) performance to MBAs trained with both MSE loss and the proposed SGD jittering, and shows stronger robustness than standard MSE-trained MBAs. However, it underperforms in OOD generalization. While DDPM serves as a strong baseline for robustness and ID accuracy, SGD jittering achieves better generalization under distribution shifts. We will add the comparison to DM in the main manuscript as an interesting baseline.
>
> It is also worth noting that DDPM requires ~10× more parameters and is significantly more data-intensive, whereas MBAs are more data-efficient [2] due to their optimization-inspired iterative structure. We also refer to prior work [1], which compares DDPM, DiffRecon, AdaDiff to MSE-trained MBAs for MRI reconstruction. Their results show that while DM can generalize well in some cases, MBA methods consistently perform better on ID data. Results in [1] shows that **MSE-trained MBA is a strong baseline**, and our proposed SGD jittering further improves their robustness and generalization.
>
> |    PSNR/SSIM     | ID | Adv. Attack | OOD |
> |:--------------------:|:-------------:|:-------------:|:-------------:|
> |     MSE training     | <28.21> / 0.603 | 25.68 / 0.382 | 29.92 / 0.779 |
> |          AT          | 27.68 / 0.564 | **27.17** / <0.549> |  7.74 / 0.597 |
> |    Input Jittering   | 28.18 / 0.595 | 25.05 / 0.420 | <29.97> / 0.740 |
> | SGD jittering (Ours) | **28.22** / <0.607> | 26.77 / **0.552** | **30.36 / 0.788** |
> |  DDPM (new baseline) | 28.17 / **0.611** | <27.13> / 0.536 | 29.72 / <0.782> |
> Table 1: MRI reconstruction. Best performances in **bold**, second best in <...>.
>
> > Robustness and generalization tradeoff
>
> We thank the reviewer for raising this important question regarding a fundamental challenge. The tradeoff has been studied from various perspectives (i.e, distributional and optimization) [3,4], with evidence that no single training objective can simultaneously optimize both. In our work, we contextualize this tradeoff within IPs by explicitly defining robustness and generalization accuracy in relation to the forward model. Our theoretical and empirical results show that AT and standard MSE training prioritize robustness and accuracy, respectively, but fail to address both objectives effectively. In contrast, the proposed SGD jittering implicitly regularizes the model, enabling simultaneous improvement in both metrics.
>
> > How to interpret $x_g$
>
> For robustness (Eq.2), $g$ is interpreted as artifacts due to noise to measurement $y$. Eq.2 measures how reconstruction $H_\theta(y_g)$ deviates from the clean $x$.
>
> For generalization (Eq.3), $x_g$ is considered as natural shift of $x$. Eq.3 measures how well the model reconstruct $x_g$ from its corresponding measurement $y_g$, or maintaining consistency with the physics model.
>
> [1] Dar et al. Adaptive diffusion priors for accelerated MRI reconstruction
>
> [2] Monga et al. Algorithm Unrolling: Interpretable, Efficient Deep Learning for Signal and Image Processing
>
> [3] Zhang et al. Theoretically Principled Trade-off between Robustness and Accuracy
>
> [4] Krainovic et al. Learning Provably Robust Estimators for Inverse Problems via Jittering
>
> We sincerely appreciate the reviewer’s constructive feedback. We believe the added experiments and clarification strengthened the manuscript and hope they satisfactorily address your concerns.

---

> > ### Comment · Reviewer_v9Xr · 2025-04-04
> >
> > I thank the authors for their rebuttal and I am raising my score to 3.

---

### Official Review · Reviewer_KSe9 · 2025-03-26

**Overall Recommendation:** 3

**Summary:**

The paper investigates robustness-accuracy tradeoffs, where the authors focus on unrolling-based methods. The authors consider different training strategies for increasing the robustness to average-case perturbations or distribution-shifts. As a specific solution for unrolling-based methods, the authors propose to add jittering noise in each step of unrolling, and demonstrate a good robustness-accuracy tradeoff for practical setups.

## update after rebuttal
Thanks again to the authors for their rebuttal. With the exception of my comment regarding the choice of noise levels, the authors addressed my concerns well. I think this is an interesting paper and keep my score as (weak) accept.

**Claims And Evidence:**

- The main claim of the paper is that SGD jittering (the proposed training strategy) yields better generalization and higher average-case robustness compared to standard MSE training.  This is not surprising, since the same holds true for other robustness-enhancing methods (input jittering or adversarial training). The authors support this by proving it for denoising setup (and technical assumptions), and provide sufficiently convincing experiments.
- Another claim of the paper is that SGD jittering improves accuracy at the same time as robustness, and thereby overcomes or mitigates the robustness-accuracy-tradeoff (see e.g. L061-066). This is very interesting, but not supported by theory, and I have some concerns and questions regarding the experimental setup.

**Essential References Not Discussed:**

NA

**Experimental Designs Or Analyses:**

I find the visualization of the toy problem results interesting, but think that real-world dataset examples would be more beneficial for the reader. The MRI results are relatively convincing, the seismic deconvolution problem is more niche. For the MRI results (and also the toy problem), I have some concerns regarding the in-distribution performance of standard MSE training. Specifically, it is very surprising to me that jittering improves the in-distribution accuracy compared to MSE training (which optimizes for it) (see also questions).
Moreover, the authors write that the jittering levels (of SGD and input methods) are chosen based on 'robustness and accuracy' (L.831) and only states the concrete values, which are hard to interpret. Since the main conclusion is that the method performs better in robustness and accuracy than competing methods, I think there should be a more principled approach to choosing it (or description of it if applicable).

**Methods And Evaluation Criteria:**

The authors train unrolled networks with the four different training strategies (MSE, adversarial training, input jittering, SGD jittering) and evaluate on in-distribution data, adversarial attacks and out-of-distribution examples (Tables 1 and 2), which is appropriate to investigate the claims. See also experimental design and analyses.

**Other Comments Or Suggestions:**

- I find the title a bit miss-leading, as "unrolling methods" (considered in the paper) is only a small subset of "model-based architectures". Moreover, the strengths of the paper are with respect to robustness, but there is noticeably less support for the accuracy part (minor comment).
- I am a bit confused by the notion of generalization, and I think the papers actually discusses robustness to distribution-shifts. While there are varying notions in the literature, generalization results are often associated to a finite sample case and in-distribution data, but here the authors effectively investigate a distribution shift by adding noise to the training distribution.
- L. 758,759: A term is missing in the inner product.
- Notation of the robustness risk: The authors denote the average-case robustness risk by $R_{\epsilon}$ and the worst-case robustness risk by $R_{e}$, which leads to confusion at first I think.
- L: 173,174: The authors write that AT learns to 'ignore' $A^{-1} e$, but this is not true in general (e.g. only when the perturbations are very large). AT finds a tradeoff between reconstructing the signal and minimizing the error.
- L. 355,356: "tradeoff in resolution"

**Other Strengths And Weaknesses:**

The investigated problem of addressing the robustness-accuracy tradeoff is important, and proposing methods which are less expensive than adversarial training is valuable. The idea of injecting noise for increasing robustness to the input, or the intermediate layers, has already been presented in the literature. However, the analysis and theoretical arguments of this approach with respect to average-case robustness is interesting and novel.

**Questions For Authors:**

- In L. 167, the authors write that AT is slow and requires iterative solvers for the attack vector. How many steps were used in PGD for adversarial training / testing?
- Regarding L. 831, could you give more details on how the jittering noise levels were chosen?
- In Figure 1 I would expect that the MSE training yields a better in-distribution results (particularly compared to adversarial training)?

**Relation To Broader Scientific Literature:**

The authors sufficiently describe the related work, including existing demonstration of robustness-enhancing methods via noise injection.

**Theoretical Claims:**

I checked the proof of Theorem 7.5 (the average-case robustness result) and don't see any major issues.

---

> ### Author Rebuttal · Authors · 2025-03-31
>
> We thank the reviewer for the thoughtful feedback and helpful suggestions. We address the reviewer’s comments and questions below.
>
> >Jittering outperforms MSE training in in-distribution results
>
> We acknowledge the reviewer’s observation and appreciate the opportunity to elaborate further. As noted in L.310–313 (prior to the Seismic Deconvolution section), training with standard MSE loss may lead to suboptimal solutions due to the highly non-convex loss surface with respect to network parameters. Our proposed SGD jittering acts similarly to layer-wise noise injection, which is known to help escape local minima by promoting exploration of the loss landscape [1]. Consequently, jittering can achieve better in-distribution and generalization performance than MSE training in some cases, as supported by both our empirical results and prior work [2].
>
> > L.167: number of steps used in PGD for AT
>
> As detailed in Appendix G (L.836), we used 20 PGD steps with a step size of 0.1 for both seismic deconvolution and MRI reconstruction tasks. While the runtime efficiency is a useful byproduct, our main point is that methods like AT and input noise injection introduce perturbations directly to the input, often break forward model consistency and result in overly smoothed reconstructions. Our contribution is to introduce SGD jittering, which promotes robustness in a more principled manner while preserving high accuracy and physical model fidelity.
>
> > How jittering noise levels were chosen
>
> As shown in Figure 5 (L.420), we compared robustness and accuracy across a range of jittering noise levels and selected the setting with the best overall trade-off. A similar hyperparameter search was conducted for the input injection baseline. Notably, we observed that input injection exhibited greater sensitivity to noise level variations, whereas our proposed SGD jittering showed more stable performance across a range of noise levels. While some tuning is still required, this suggests that SGD jittering is more robust to hyperparameter selection in practice. We will include a short discussion of this in the main paper.
>
> > Notion of Generalization
>
> We agree that in classical learning theory, generalization typically refers to a model's ability to perform well on unseen in-distribution data, often analyzed in the finite-sample setting. However, in this work, we adopt a broader and increasingly common notion of generalization under distribution shift, which aligns with recent literature on robustness and transferability. To avoid confusion, we will revise the manuscript to clarify this broader usage of "generalization" and explicitly distinguish it from classical in-distribution generalization.
>
> > Typos and clarification
>
> We thank the reviewer to pointing it out, we will fix the typo and make clarification of the terms in the manuscript.
>
> [1] Orvieto et al. Explicit Regularization in Over-parametrized Models via Noise Injection
>
> [2] Lim et al. Noisy Recurrent Neural Networks
>
> We sincerely appreciate the reviewer’s time for reviewing and the constructive suggestions, and we believe that the additional clarifications improve the quality of the submission. We hope this addresses your concerns.

---

> > ### Comment · Reviewer_KSe9 · 2025-04-08
> >
> > Many thanks to the authors for their rebuttal and for addressing my concerns. The paper is interesting and I keep my score as (weak) accept.
> >
> > One comment further to the choice of noise levels: The rebuttal states that SGD jittering is less sensitive to the choice compared to input jittering. Thats an interesting observation, but the systematic way to choose the final noise levels should be clearly stated (beyond "best robustness-accuracy"), since these choices are important for baselines (e.g. input jittering). I encourage to include this in the dicussion as well and add more details in the plots (e.g. Figure 5) regarding noise levels or adversarial attack levels $\epsilon$.

---

### Decision · Program_Chairs · 2025-05-01

**Decision:**

Accept (poster)

**Comment:**

The paper considers model based architectures for solving inverse problems, and proposes a training strategy which inects noise at each iteration during SGD, called SGD jittering. The paper shows that jittering can generalize better and can be more robust to average case perturbations.

The paper received four reviews and after rebuttal, all are supportive of publication.

The reviewers find the paper's theoretical and empirical results interesting.

The reviewers identified some minior weaknesses, including:
- Reviewer v9Xr finds the main weakness is the lack of baselines beyond unrolled networks; the authors added a Diffusion model as baseline.
- Reviewer mUA8 notes that is would be interesting to add results beyond denoising, e.g., for compressive sensing. I agree with the reviewer that this would be interesting and with the authors that this is beyond of the scope of the paper.
- Reviewer KSe9 had some concerns regarding the statement that SGD jittering improves accuracy at the same time as robustness.

I also find the paper interesting and recommend it for acceptance.

Besides the feedback from the reviewers, in order to understand the robustness benefits of the method better I think it would be interesting (in the final version or follow-up work) to evaluate out-of-distribution performance as a function of in-distribution performance (see e.g., Fig. 2 in Darestani et al. ``Measuring Robustness in Deep Learning Based Compressive Sensing’’, ICML 2021). By evaluating out-of-distribution performance like this it becomes clear whether the improved performance to out-of-distribution data (like in Figure 4b in the paper under review) is due to a robustness benefit or is because the in- and out-of-distribution performance improves (i.e., the algorithm still lies on a line as in Fig 2 in Darestani et al).